# Synthesis of $CuCo_2S_4$@Expanded Graphite with crystal/amorphous heterointerface and defects for electromagnetic wave absorption

Zhimeng Tang[1,2,3], Lei Xu[1,2,3,4] ✉, Cheng Xie[1,2,4], Lirong Guo[2,3,4], Libo Zhang[1,2,4], Shenghui Guo[1,2,4] & Jinhui Peng[1,2,4] ✉

The remarkable advantages of heterointerface and defect engineering and their unique electromagnetic characteristics inject infinite vitality into the design of advanced carbon-matrix electromagnetic wave absorbers. However, understanding the interface and dipole effects based on microscopic and macroscopic perspectives, rather than semi-empirical rules, can facilitate the design of heterointerfaces and defects to adjust the impedance matching and electromagnetic wave absorption of the material, which is currently lacking. Herein, $CuCo_2S_4$@Expanded Graphite heterostructure with multiple hetero-interfaces and cation defects are reported, and the morphology, interfaces and defects of component are regulated by varying the concentration of metal ions. The results show that the 3D flower-honeycomb morphology, the crystal-crystal/amorphous heterointerfaces and the abundant cation defects can effectively adjust the conductive and polarization losses, achieve the impedance matching balance of carbon materials, and improve the absorption of electromagnetic wave. For the sample CEG-6, the effective absorption of Ku band with $RL_{min}$ of −72.28 dB and effective absorption bandwidth of 4.14 GHz is realized at 1.4 mm, while the filler loading is only 7.0 wt. %. This article reports on the establishment of potential relationship between crystal-crystal/ amorphous heterointerfaces, cation defects, and the impedance matching of carbon materials.

The rapid development of advanced electronic equipment and wire-less communication technology has brought great convenience to people's life, but it also produces electromagnetic radiation pollution that cannot be ignored. This pollution would cause serious inter-ference with the normal operation of the surrounding electronic components, reduce information security and communication quality, and endanger human health[1–3]. It is one of the most effective ways to develop advanced electromagnetic wave (EMW) absorbing materials

to solve this problem, and these materials usually demonstrate fea-tures of lightweight, strong absorption, thin thickness, and wide band width[4–8]. Carbon materials, such as expanded graphite (EG), graphene (GR), and carbon nanotubes (CNTs)[9–11], have good prospects for EMW absorption due to their low density, rich functional groups, and adjustable electrical properties. How to balance the impedance-matching characteristics of carbon materials with high EMW absorp-tion performance is the focus of research in this field. Currently, the

[1]Faculty of Metallurgical and Energy Engineering, Kunming University of Science and Technology, Kunming 650093, PR China. [2]National Local Joint Laboratory of Engineering Application of Microwave Energy and Equipment Technology, Kunming University of Science and Technology, Kunming 650093, PR China. [3]State Key Laboratory of Complex Nonferrous Metal Resources Clean Utilization, Kunming University of Science and Technology, Kunming 650093, PR China. [4]The Key Laboratory of Unconventional Metallurgy, Ministry of Education, Kunming University of Science and Technology, Kunming 650093, PR China. ✉e-mail: xu_lei@kust.edu.cn; jhpeng@kust.edu.cn

designed construction of heterogeneous interfaces or defects is the main route to prepare high-performance carbon-based wave-absorbing materials[12–15]. The design of heterogeneous interfaces produces a suitable complex dielectric constant of the material and facilitates the improvement of impedance matching[16–18]. Meanwhile, the construction of heterogeneous interfaces causes asymmetry of nearby electrons and crystal structure, and these defects lead to the lattice distortion[19], charge mismatch[20], and band migration[21], which are also conducive to EMW attenuation. All these measures are expected to enhance the material absorption performance.

As a three-dimensional (3D) carbon skeleton material, EG has the advantages of high thermal conductivity, high electrical conductivity, low density, low preparation cost, and easy mass production. Moreover, the unique honeycomb structure of EG is conducive to the multiple reflection and enhanced absorption of EMW. Strong EMW attenuation is achievable at a low content of EG as a filler substrate[22–24]. However, it is difficult to obtain strong EMW attenuation due to the poor impedance matching of single-component EG. Therefore, other components are usually integrated with EG to regulate the compositions, interfaces, and defects and optimize the electromagnetic parameters and absorbing properties of the materials. Currently, transition metal sulfides are of wide interest in the modification of EMW absorbing materials. Among them, copper-cobalt-based bimetallic sulfide ($CuCo_2S_4$) is a spinel-type sulfide copper-cobalt mineral. The occupation of multiple cations in the mixed spinel endows it with tunable electrical properties[25], resulting in strong dielectric loss characteristics. Moreover, the morphological changes of $CuCo_2S_4$ are beneficial to the impedance matching regulation. Therefore, $CuCo_2S_4$ modified EG was used to construct 3D carbon-based wave-absorbing materials with heterogeneous interfaces and synergistic mechanism of defect engineering, so as to promote the impedance matching and

multiple loss effect of composite materials. This advanced approach has the most fundamental influence on the conduction loss, polarization loss and magnetic response of carbon-matrix EMW absorber, so it is expected to obtain excellent microwave absorption performance.

Herein, we propose a strategy to regulate the impedance matching of carbon-matrix absorbing materials through the crystal/amorphous heterointerfaces and cation defects. The 3D flower honeycomb $CuCo_2S_4$@EG heterostructures were successfully prepared by in-situ growth of $CuCo_2S_4$ on a 3D carbon matrix (EG) by microwave solvothermal method, and the flower shape of $CuCo_2S_4$ was adjusted by the ratio of copper and cobalt atoms. This unique 3D heterogeneous structure can provide abundant heterointerfaces and defects, which can effectively adjust the impedance matching of carbon-matrix materials to achieve multiple attenuation of EMW. This work illustrates the relationship between the internal composition, structure, and function of $CuCo_2S_4$@EG EMW absorbers, and provides a simple and effective strategy for the development and design of high-performance 3D carbon-matrix absorbing materials.

## Results and discussion

The schematic diagram of the preparation process of $CuCo_2S_4$@EG composites was presented in Fig. 1. In the study, EG was used to construct 3D honeycomb-like carbon-matrix conductive network backbones. A thorough investigation of the structural and physical characteristics of EG before and after microwave treatment were measured (Supplementary Figs. 1–3). Compared to the general Hummer method, the microwave-prepared EG showed better electrical conductivity and excellent thermal conductivity (Supplementary Table 1). Meanwhile, the unique honeycomb structure of EG helped create a perfect conductive network and achieve the multiple reflections of EMW[26]. In addition, $CuCo_2S_4$ nanoflower arrays were rapidly

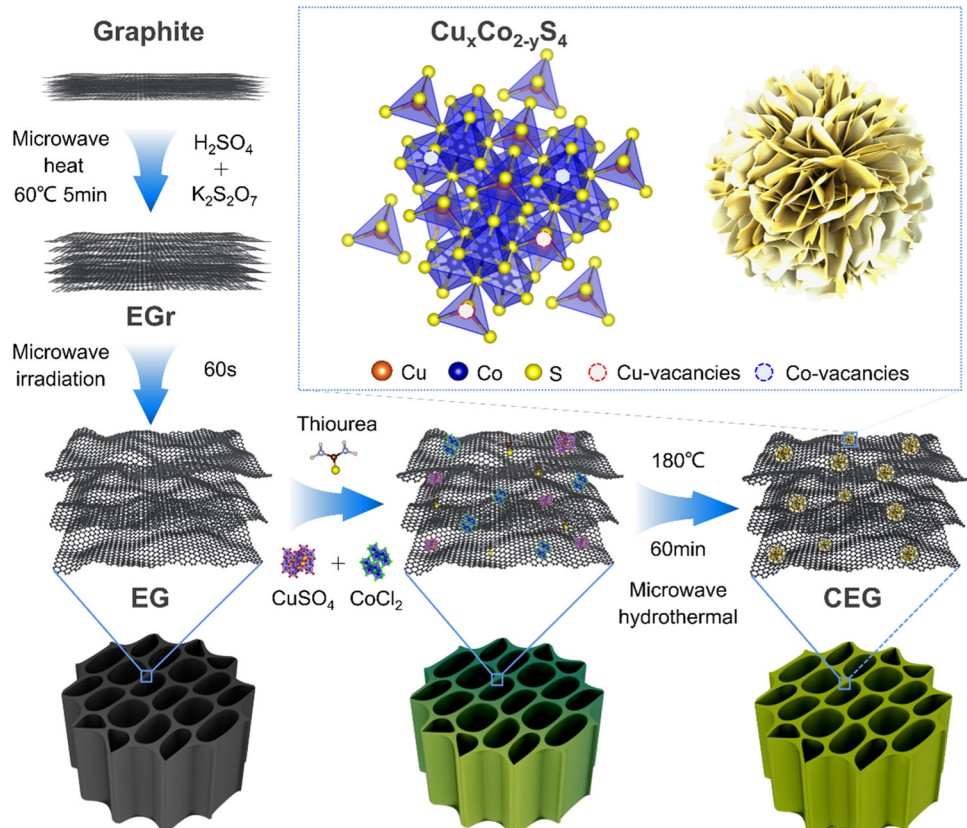

**Fig. 1 | Schematic diagram of preparation and microstructure of $CuCo_2S_4$@EG heterostructures.** Construction of heterostructures through in-situ growth of defect-rich $CuCo_2S_4$ on expanded graphite.

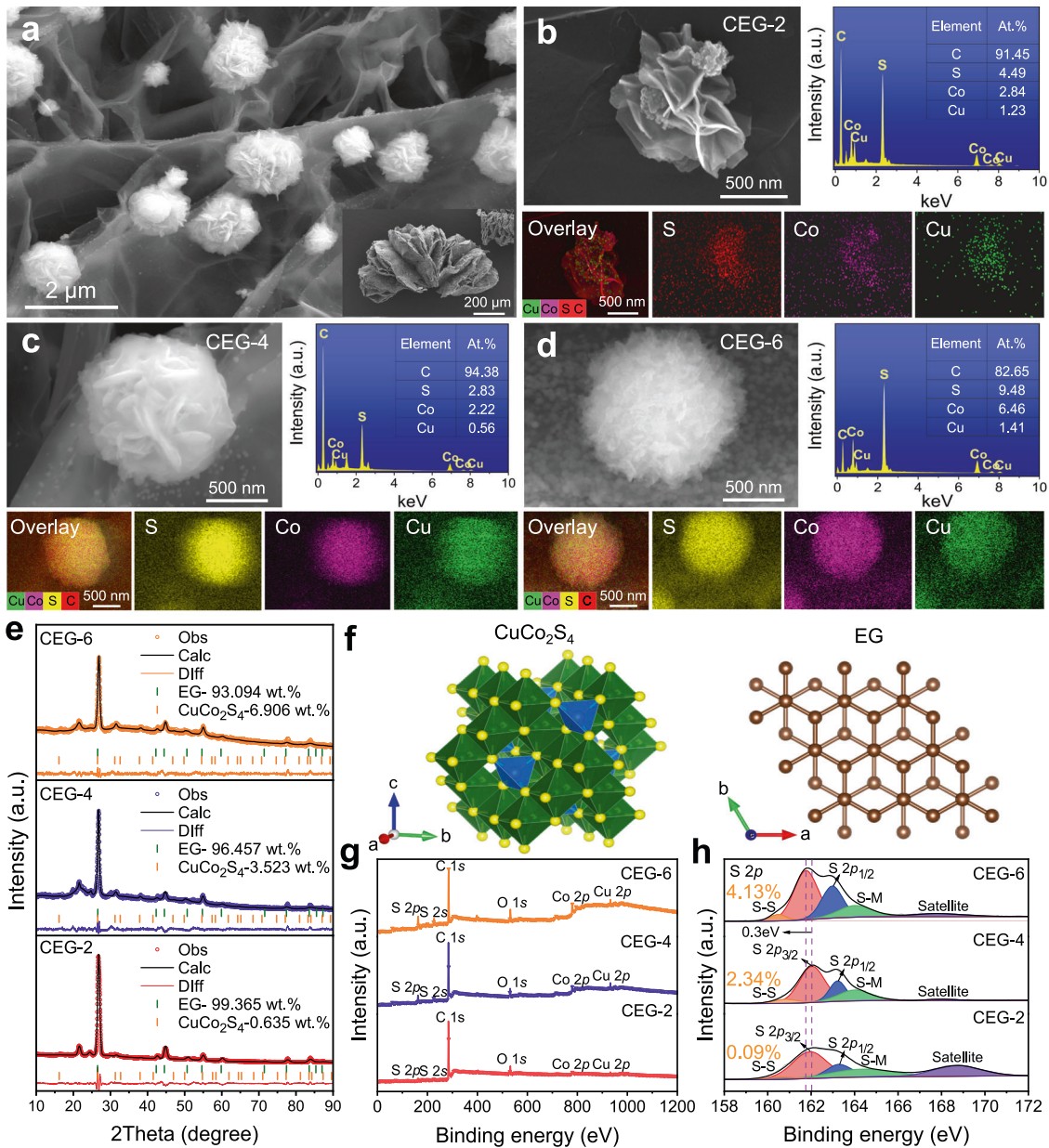

**Fig. 2 | Morphological and compositional characterization of CuCo₂S₄@EG heterostructures. a** Microscopic morphology of CuCo₂S₄@EG. **b–d** Microscopic morphology and corresponding element distribution of CEG-2, CEG-4 and CEG-6. **e** Phase composition and content of CEG-2, CEG-4 and CEG-6. **f** Crystal structure of CuCo₂S₄ and EG. **g** XPS spectra of CEG-2, CEG-4 and CEG-6, and **h** S 2p spectra of CEG-2, CEG-4 and CEG-6, the yellow number indicates the S-S ratio.

grown in situ on the 3D honeycomb EG by the microwave solvothermal method, resulting in the flower-honeycomb CuCo₂S₄@EG heterostructure. It is expected that this unique flower-honeycomb heterostructure is beneficial to enhance the multiple reflections and interfacial polarization loss of EMW.

The microscopic morphology of CuCo₂S₄@EG EMW absorber was observed by scanning electron microscope (SEM), as shown in Fig. 2. A characteristic flower-like structure was developed on the surface of EG (Fig. 2a and Supplementary Fig. 4). This distinctive 3D flower-like structure facilitated the good matching of the impedance of the composites and the effective attenuation of EMW[27,28]. On this basis, CuCo₂S₄@EG heterostructures with different flower-like morphologies were prepared by regulating the atomic ratio of Co to Cu (Co: Cu = 2:1, 4:1, 6:1, and 8:1), which were labeled as CEG-X (X = 2, 4, 6, 8), as shown in Supplementary Figs. 5 and 6. CuCo₂S₄ exhibited a sheet-based petal-like structure at the Co/Cu atomic ratio of 2:1 (Fig. 2b). The

increase in the number of Co atoms led to the less obvious lamellar structure and significant changes in the petal-like structure of CuCo₂S₄. With the gradually increasing number of petals, the typical petal-like structure was changed to a flower floccule structure (Fig. 2c, d). When the atomic ratio of Co to Cu was 8:1, CuCo₂S₄ presented a disordered spherical shape (Supplementary Fig. 7).

The specific surface area and pore size distribution of CEG composites were obtained, as shown in Supplementary Fig. 8a–c. Some micro-medium-large pores were observed for CEG composites. The specific surface area and pore volume of EG are 34.8 m²g⁻¹ and 0.17 cm³g⁻¹, respectively. With the introduction of CuCo₂S₄, the specific surface area and pore volume of the composite decreased to only 1.0 m²g⁻¹ and 0.02 cm³g⁻¹ for CEG-2. The increase of Co/Cu ratio resulted in the improved specific surface area and pore volume, due to the formation of the petal-like structure of the composite. The specific surface area and pore

volume of CEG-6 reached 13.7 $m^2g^{-1}$ and 0.03 $cm^3g^{-1}$. The multilevel pore structure of composite material was beneficial to improve the impedance matching[29].

The crystal phase composition of the samples was analyzed using X-ray diffraction (XRD, Supplementary Fig. 9). It was interesting that the samples with different Co/Cu ratios, such as CEG-2, CEG-4, and CEG-6, all showed similar peak positions, which could be well ascribed to the $CuCo_2S_4$ phase (JCPDS No. 42-1450). To further clarify the detailed phase and crystallographic information of CEG hetero-structures, the Rietveld refined analysis method, and TOPAS software were used to analyze the crystal structure and phase content in the XRD pattern (Fig. 2e, f). According to the findings, the $CuCo_2S_4$ phase in all samples showed spinel structure (Fd-3m), but the $CuCo_2S_4$ phase content was gradually improved with the enhancing cobalt atomic concentration.

Meanwhile, the molecular structure and chemical state of CEG heterostructure surface were analyzed by the X-ray photoelectron spectroscopy (XPS). As shown in Fig. 2g, the characteristic peaks of S, C, O, Co, and Cu could be observed in the range of 0-1200 eV. The introduction of 3D flower-like $CuCo_2S_4$ phase did not cause the change of carbon and oxygen functional groups on the surface of EG conductive network (Supplementary Fig. 10a, b). The peaks at 284.8, 285.1, and 286.6 eV were attributed to C-C, C-O, and C = O[26,30], respectively. The two prominent main peaks at 931.9 eV and 951.6 eV (Supplementary Fig. 10c) were ascribed to the Cu $2p_{3/2}$ and Cu $2p_{1/2}$ spin orbitals, respectively, indicating the existence of Cu ions in mixed valence states of +1 and +2[31]. However, as the Co/Cu ratio increased, the peak corresponding to Cu $2p_{3/2}$ moved to higher binding energy regions ($\approx 0.21$ eV). In addition, the same results were observed in the high-resolution spectrum of Co $2p$ (Supplementary Fig. 10d), which proved the electron-donating role of Cu and Co[32,33].

In the XPS spectra of S $2p$ (Fig. 2h), two prominent peaks (at 161.8 eV and 163.1 eV) and a satellite peak at 164.1 eV, which represented a typical metal-sulfur bonding, were observed. However, as the Co/Cu ratio rose, the peak corresponding to S $2p_{3/2}$ was shifted to lower binding energy regions ($\approx 0.3$ eV), indicating the transfer of electrons from the Co and Cu sites to the S sites and the generation of strong electron interactions. It was previously reported that strong electron interactions can easily promote the production of severely surface-disordered structures and a large number of defect levels in the transition metal sulfides[12,19,34]. Moreover, it was particularly note-worthy that a new characteristic peak belonging to the S-S bond was observed at 160.5 eV[19,35], which was caused by the Cu or Co cation defects in CEG.

The detailed microstructure and lattice information of the CEG were studied using high-resolution transmission electron microscopy (HRTEM) and aberration-corrected transmission electron microscopy (AC-TEM). Obvious heterointerfaces in the CEG (Fig. 3a and Supplementary Figure 11), were conducive to regulating the electrical conductivity[36] and promoting the interface polarization. Meanwhile, the measured lattice fringes of 0.337 nm and 0.285 nm can be indexed to the (002) plane of EG and the (113) plane of $CuCo_2S_4$, respectively. Especially, it should be noted that significantly large amounts of lattice defects and lattice distortions were present in the $CuCo_2S_4$ and EG lattices (Fig. 3b and Supplementary Fig. 12). The amorphous structure resulted in the generation of a significant number of discontinuous lattice fringes (Fig. 3b2 and Supplementary Fig. 12b). The lattice distortions, and discontinuous lattice fringes facilitated the disruption of the lattice symmetry in $CuCo_2S_4$@EG, leading to the lattice mismatch, and variations of the electron distribution state and charge transfer path, thereby promoting the interfacial polarization. Moreover, the presence of a large number of vacancies in $CuCo_2S_4$ and EG was observed (Fig. 3c, e). No S vacancy characteristic peak in $CuCo_2S_4$ was found in the electron paramagnetic resonance (EPR) (Fig. 3f), which

further confirmed the speculation that there was Cu or Co cation defects in CEG.

To further analyze the coordination environment of Cu and Co atoms, X-ray absorption fine structure (XAFS) analysis was performed. As shown in the X-ray absorption near edge spectra (XANES) (Fig. 3g and Supplementary Fig. 13a), the fitting results exhibited that the valence states of Cu and Co in CEG were +2 and +3, respectively, which was consistent with the results of XPS[31,37]. The Fourier transform (FT) of the $k^3$-weighted extended X-ray absorption fine structure (EXAFS) spectra of CEG displayed one main peak at about 1.8 Å (Fig. 3h and Supplementary Fig. 13b), which was assigned to the Cu-S and Co-S bonds. To gain more information about the atomic configuration of the Cu and Co species in CEG-6, the wavelet transform (WT)-EXAFS analysis was performed. The WT signals of the Cu-Cu, Cu-O, and Co-Co bonds were not detected in CEG (Fig. 3i and Supplementary Fig. 13e). There was only one maximum intensity in WT contour maps of Cu and Co, which was about 5.5 Å and 8.0 Å, corresponding to the coordination of Cu-S and Co-S, respectively. The fitting results of CEG in R and k spaces are shown in Fig. 3j, k and Supplementary Fig. 13c, d, respectively. The coordination numbers of Cu-S and Co-S in CEG were about 3.2 and 2.9, and the average bond lengths of Cu-S and Co-S were 2.28 Å and 2.26 Å, respectively (Supplementary Table 2). The above analysis demonstrated that each Cu and Co atom in CEG was coordinated with about 3 S atoms, which was lower than $CuCo_2S_4$ (Fd-3m) standard structure. AC-TEM, EPR, XPS, and XAFS analyses indicated the exis-tence of Cu and Co cation defects in CEG. In addition, the concentra-tion of Cu and Co cation vacancy in CEG was enhanced with rising Co/Cu ratio (Fig. 3l). In general, the presence of such metal cation defects would disrupt the charge balance and affect electron transport within the material[38].

In order to study the electronic structure changes caused by heterostructure and cation defects in CEG, the density of states (DOS) and partial density of states (PDOS) for CEG heterogeneous structures with cation defects were calculated by means of density functional theory (DFT) (Supplementary Fig. 14). Compared with $CuCo_2S_4$, the energy bands of the CEG heterostructure were shifted, and the Fermi energy level was reduced, in company with the appearance of peaks corresponding to EG (Supplementary Fig. S15). In addition, as shown in Supplementary Fig. 16, new electronic states (p bands) were formed near the valence band and Fermi level due to the presence of Cu-Co vacancy, indicating that the construction of cation defects in CEG was beneficial to improve its conductivity[39]. The electrical conductivity of EG was relatively high (0.74 S·m$^{-1}$, Supplementary Fig. 17) due to the "skin effect", and the high conductivity was not conducive to impe-dance matching[40]. The conductivity of $CuCo_2S_4$ was significantly lower than that of EG, and demonstrated a rising trend with the increase of Co-Cu ratio.

The electromagnetic parameters of the $CuCo_2S_4$@EG composite with a filling volume of 7.0 wt.% was measured by Vector network analyzer (VNA) in the frequency domain of 1 - 18 GHz (Fig. 4a–d). And the value of $C_0$ was calculated (Supplementary formula (1)) As shown in Supplementary Fig. 18, the $C_0$ value of EG presented natural resonance phenomenon at 1-4 GHz, while the tendency constant at 4-18 GHz presented eddy current loss characteristics[41–43]. In addition, the mag-netic properties of EG and CEG composites were tested at room tem-perature. EG showed weak hysteresis characteristics, while CEG composites showed no obvious hysteresis phenomenon (Supple-mentary Fig. 19a). And $\mu'$ and $\mu''$ were close to 1 and 0 (Fig. 4a, b). This indicated that the magnetic loss contributed little to the attenuation absorption of EMW in CEG composites. Therefore, the EMW absorp-tion performance of CEG composites was regulated mainly through dielectric loss.

EG displayed a high dielectric constant and dielectric loss tangent (Fig. 4c–e). After the introduction of flower-like $CuCo_2S_4$, the dielectric

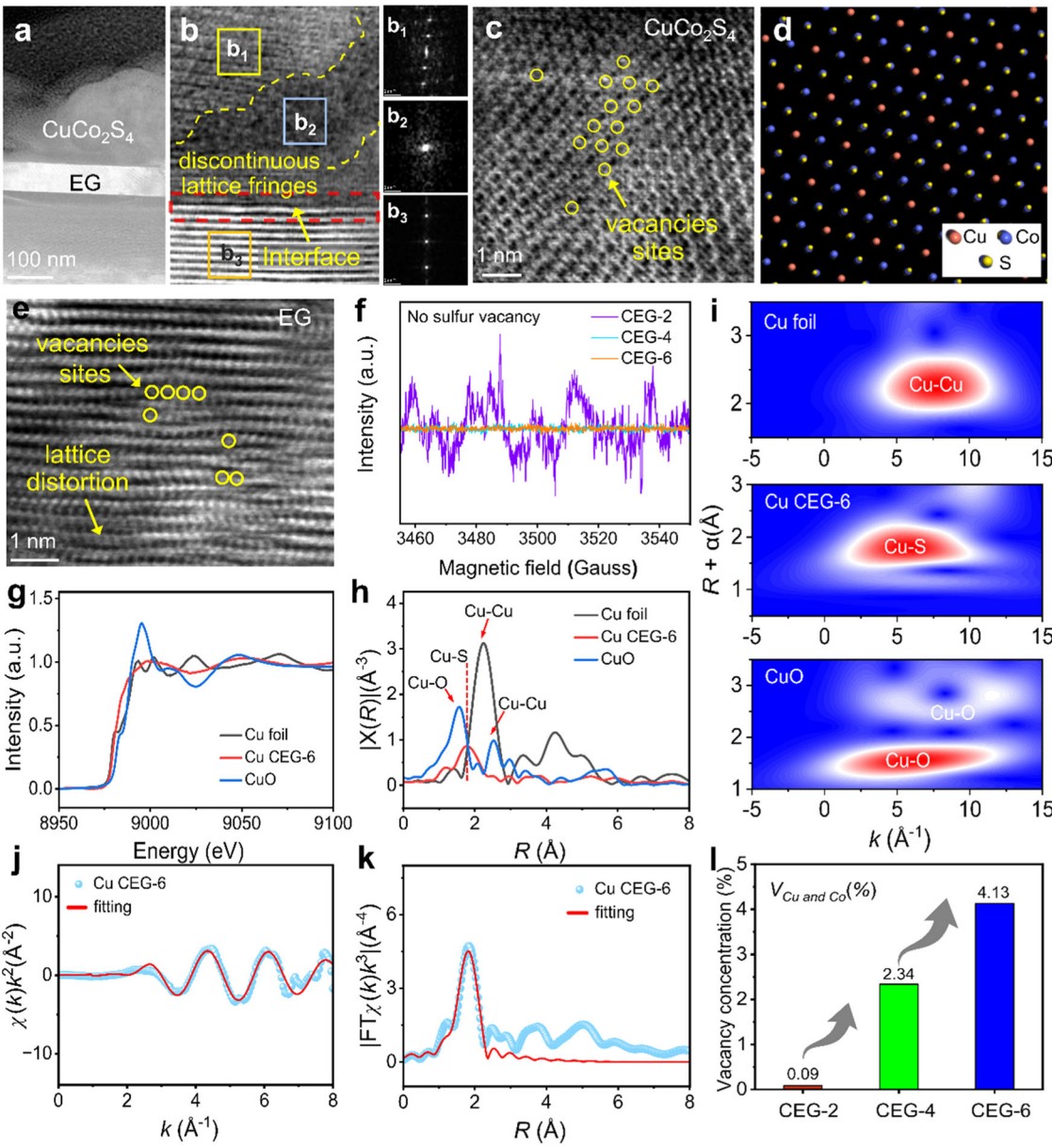

**Fig. 3 | Microstructure and atomic coordination information of the CuCo₂S₄@EG. a**, **b** AC-TEM micrograph of CEG-6. **c** HAADF-STEM micrograph of CuCo₂S₄. **d** Distribution diagram of CuCo₂S₄. **e** HAADF-STEM micrograph of EG. **f** EPR spectrum of CEG-2, CEG-4 and CEG-6. **g** Normalized XANES spectra at the Cu K-edge of the Cu foil, CEG-6 and CuO. **h** FT-EXAFS spectra of the Cu foil, CEG-6 and CuO. **i** WT of the Cu foil, CEG-6 and CuO. **j** EXAFS fitting curve for Cu CEG-6 in the k-space. **k** EXAFS fitting curve for Cu CEG-6 in the R-space. **l** Vacancy concentration of CEG-2, CEG-4 and CEG-6 samples.

constant and dielectric loss tangent decreased significantly. However, the dielectric constant and dielectric loss tangent tended to improve with the enhancement of Co/Cu ratio (Supplementary Fig. 20). In order to analyze the dielectric loss mechanism of CEG, Cole-Cole curve was drawn according to Debye theory and Supplementary formula (2)[44], as shown in Supplementary Figs. 21a2-d2 and 22. The Cole-Cole curve of EG had long straight lines and nine small semicircles. This may be attributed to the $\varepsilon_c''$ and $\varepsilon_p''$. (Supplementary Fig. 22). The mesh skeleton of EG had good electrical conductivity, which was conducive to the generation of conduction loss. Lattice defects in EG were conducive to trigger the dipole polarization under the action of electromagnetic wave, resulting in polarization loss. Compared with EG, the number of semicircles in CEG enhanced with distorted shape, displaying the mechanism of multiple polarization. This may be because the heterogeneous interface in CEG greatly hindered the charge flow[45], resulting in uneven distribution of electrons at the interface and promoting the

interface polarization. In addition, cation defects in CuCo₂S₄ components were favorable for dipole polarization. With the rising Co/Cu, the cation vacancy concentration in CEG was improved, further promoting the polarization loss.

In order to further analyze the influence of $\varepsilon_p''$ and $\varepsilon_c''$ on dielectric loss in CEG, the electrical conductivity of the samples was tested (Supplementary Figure 23), and $\varepsilon_p''$ and $\varepsilon_c''$[46] were obtained according to Debye's theory and Supplementary Equations (3), (4) and (5), (Fig. 4f and Supplementary Fig. S21). The conduction loss was dominant below 2 GHz, and the polarization loss was dominant at 2–18 GHz. In addition, the $\varepsilon_p''$ value of CEG increased with rising Co/Cu, which indicated enhanced polarization loss. Therefore, heterojunction and cation defects had important effects on dielectric loss.

The difference of charge density in the heterogeneous interface between CuCo₂S₄ and EG was analyzed by DFT calculation (Supplementary Fig. 24). The charge was redistributed at the interface of

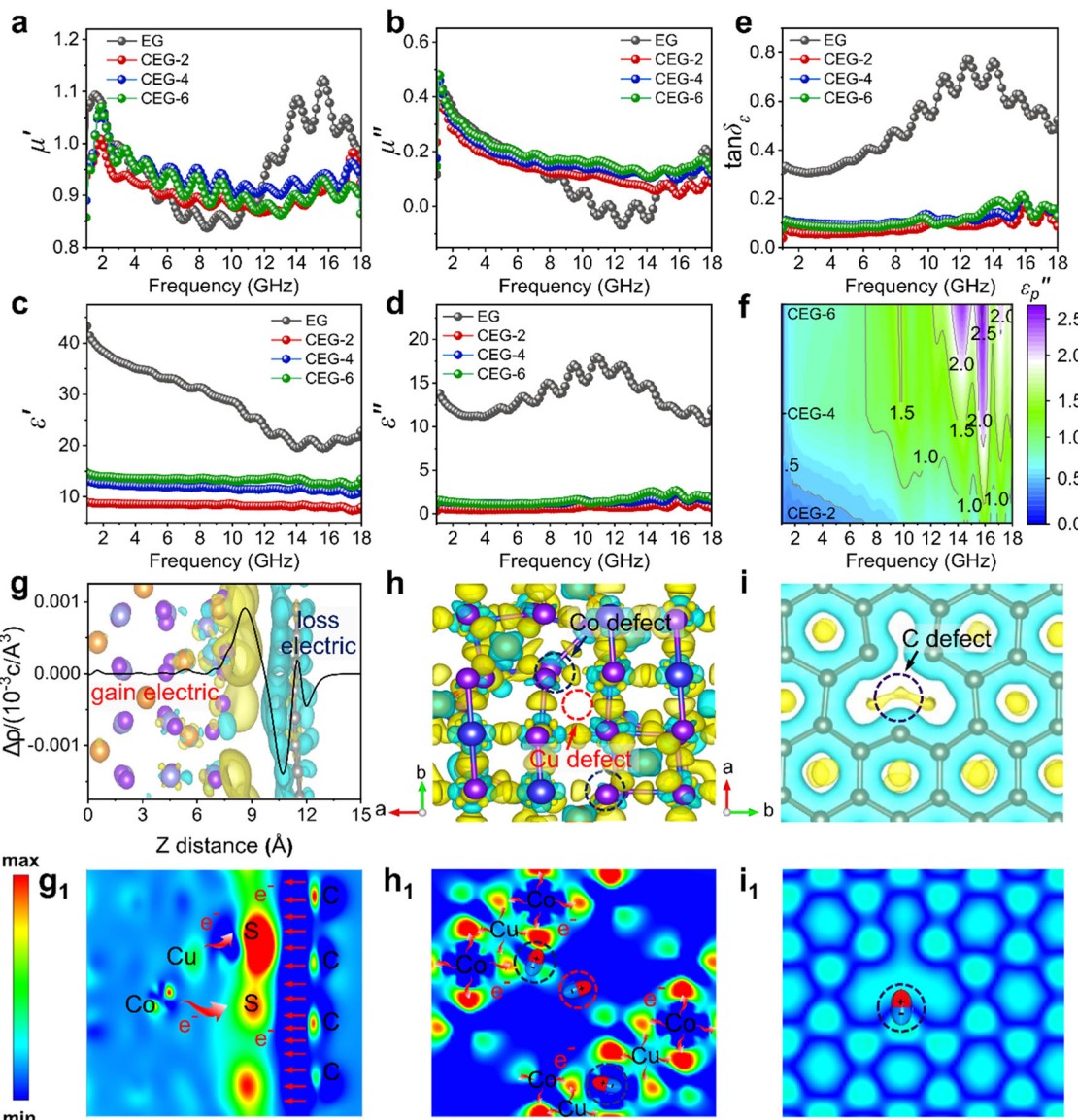

**Fig. 4 | Electromagnetic wave absorption properties of CuCo₂S₄@EG. a, b** Real part and imaginary part of complex permeability of EG, CEG-2, CEG-4 and CEG-6. **c, d** Real part and imaginary part of complex permittivity of EG, CEG-2, CEG-4 and CEG-6. **e** Dielectric loss tangent (tan $\delta_\varepsilon$) of EG, CEG-2, CEG-4 and CEG-6. **f** $\varepsilon_p''$ of CEG- 2, CEG-4 and CEG-6. **g, g₁** The charge density difference of heterointerface of CuCo₂S₄@EG. **h, h₁** The charge density difference of defect CuCo₂S₄. **i, i₁** The charge density difference of nonperfect EG. Blue-green color represents charge depletion, while yellow color represents charge accumulation.

CuCo₂S₄ and EG, which resulted in electron accumulation at the interface of CuCo₂S₄ and electron depletion at the interface of EG (Fig. 4g, g1). The uneven distribution of positive and negative charges in space could promote interfacial polarization and thus enhance the dissipation of EMW. Meanwhile, the charge distribution of EG and CuCo₂S₄ phase defects was calculated. Due to the absence of C, Cu, and Co, the negative and positive charge centers at the defect were redistributed and a local electric field was established to form a permanent dipole (Fig. 4h, h1 and Fig. 4i, i1). These permanent dipoles oscillated in an external electromagnetic field, promoting dipole polarization loss. In addition, with increasing Co/Cu ratio, the concentration of metal cation vacancy in CEG increased. The further increase of dipoles in CEG promoted the polarization loss, $\varepsilon_p''$, and the attenuation of EMW energy.

Subsequently, the EMW absorbing properties of the CuCo₂S₄@EG samples were evaluated based on the transmission line theory. An optimal absorber should be capable of attenuating over 90% of the incident EMWs, with its reflection loss (RL) value lower than -10 dB, and

its corresponding frequency range was regarded as effective absorption bandwidth (EAB)[29,47]. The CEG-6 sample exhibited the strongest EMW absorbing ability with an ultra-low filler loading (7.0 wt.%) (Supplementary Fig. 25 and Fig. 5a1–d1). The RL$_{min}$ and EAB values of CGE-6 in the Ku band reached -72.28 dB and 4.14 GHz, respectively, while the thickness was only 1.4 mm. Nevertheless, when the thickness of CEG-4 sample was 1.4 mm, the RL$_{min}$ and EAB values of the composite in the Ku band were -19.29 dB and 3.12 GHz, respectively. Moreover, EG showed ultra-low EMW absorption characteristics with its minimum reflection loss (RL min) of only -7.07 dB.

In general, only when the impedance matching (Z) and the attenuation constant (α) must be appropriate, can the effective EMW absorption performance be obtained[48]. For the impedance matching, the modulus of normalized input impedance ($|Z_{in}/Z_0|$) was often selected to characterize the impedance matching condition, which can be expressed by the Supplementary Equation (6)[49].

Generally, $|Z_{in}/Z_0|$ in the regions of 0.8 and 1.2 should be considered as ideal impedance matching performances (Fig. 5a2–d2.)[23,50].

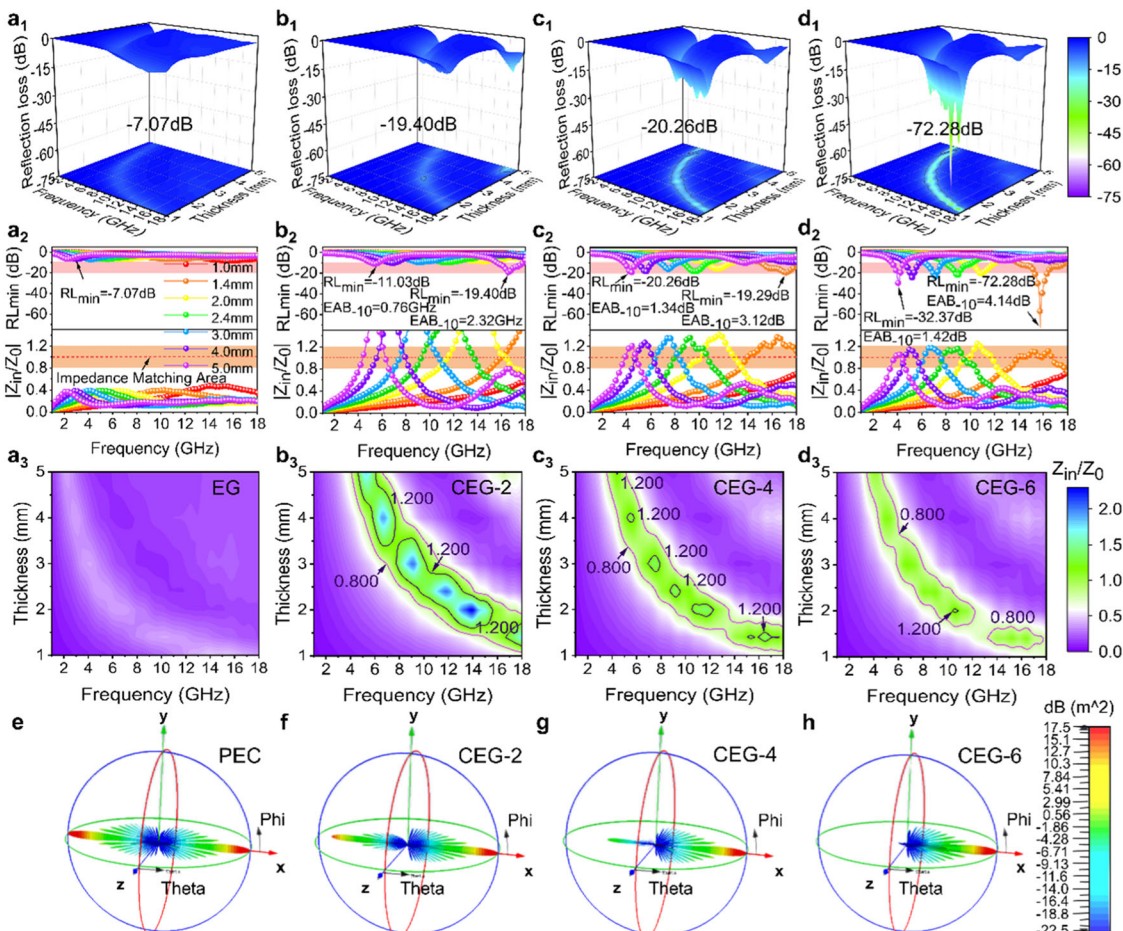

**Fig. 5 | Electromagnetic wave absorption properties of CuCo₂S₄@EG. a₁–a₃** RL maps and |Z_in/Z_0| values of EG. **b₁–b₃** RL maps and |Z_in/Z_0| values of CEG-2. **c₁–c₃** RL maps and |Z_in/Z_0| values of CEG-4. **d₁–d₃** RL maps and |Z_in/Z_0| values of CEG-6. RL maps and |Z_in/Z_0| values with different thicknesses (1.0-5.0 mm) during the frequency range of 1-18 GHz. **e–h** The far-field response based on the plane wave theory of PEC, CEG-2, CEG-4 and CEG-6 (1.4 mm).

For a clearer expression, the impedance values Z for all samples were given in Fig. 5a3–d3. Due to the high conductivity of EG, the impedance matching was poor. A large number of heterogeneous interfaces were formed by in-situ growth of nano-flower-like CuCo₂S₄ on the surface of EG, which reduced the conductivity of the sample and improved the $\left|Z_{in}/Z_0\right|$. However, the $\left|Z_{in}/Z_0\right|$ of CEG-2 and CEG-4 were both higher than 1.2, but a good impedance matching was not achieved. When the ratio of Co/Cu was 6:1, the concentration of cation vacancy in CEG was improved. The conduction loss and polarization loss were enhanced. The composite exhibited the best impedance matching. In addition, the flower-like structure and high α value of CEG-6 benefited the penetration and loss of EMW (Supplementary Fig. 26), and thus enable a better EMW absorption performance. It was worth noting that when the Co/Cu ratio reached 8:1, the CEG-8 morphology became a disordered spherical structure with decreasing defect concentration in the material. At the same time, a new phase with poor conductivity was produced. Therefore, its dielectric loss and EMW absorption performance became worse (Supplementary Figs. 27 and 28).

In addition, the 3D RCS simulation of CuCo₂S₄@EG was carried out using CST Microwave Studio to simulate the EMW absorption characteristics of the composites under realistic application conditions. The simulation model consisted of CuCo₂S₄@EG and a perfect electric conductor (PEC) (Supplementary Fig. 29a). It monitored the reflected signal intensity of the pure PEC plane and the CuCo₂S₄@EG coating under the action of a vertically incident electromagnetic wave for the entire detection angle (Supplementary Fig. 29b, c). The thickness and frequency were set at 1.4 mm and 15.75 GHz, respectively. As demonstrated in Fig. 5e-h, although the 3D RCS distribution of CuCo₂S₄@EG presented a comparable shape, the vertical reflection intensity of CEG-6 was much weaker than that of CEG-4 and CEG-2. This suggested that CEG-6 can achieve perfect absorption of EMW when the absorption layer was only 1.4 mm.

To evaluate the heat dissipation capability and environmental adaptability of the CuCo₂S₄@EG heterostructures, the heat dissipation ability of CEG-6 samples at 2.45 GHz microwave power was observed by the infrared thermography technique. The CuCo₂S₄@EG composites inherited the excellent thermal conductivity of EG (Fig. 6a). At 2.45 GHz and 300 W, the temperature of the composites rose from room temperature to 199.7 °C within 5 s. Then with the turn-off of the microwave source, synchronously the temperature fell from 199.7 to 35.8 °C in only 10 s (Fig. 6b, c). The results showed that CEG-6 composites could transform microwave energy into heat energy and dissipate it in time through the coordination of dielectric polarization (interface polarization and dipole polarization) and eddy-current loss. In addition, the performance of similar EMW absorbers was compared to highlight the excellent EMW absorption of CEG. The CEG demonstrated more outstanding EMW absorption performance and excellent heat dissipation characteristics compared to the carbon-matrix absorbing materials reported previously (Supplementary Fig. 30 and Table 3), making it more valuable in practical application[51–60].

The aforementioned discussion led us to elucidate that the impedance matching and the EMW absorption capacity of the composites could be adjusted by modifying the morphology and

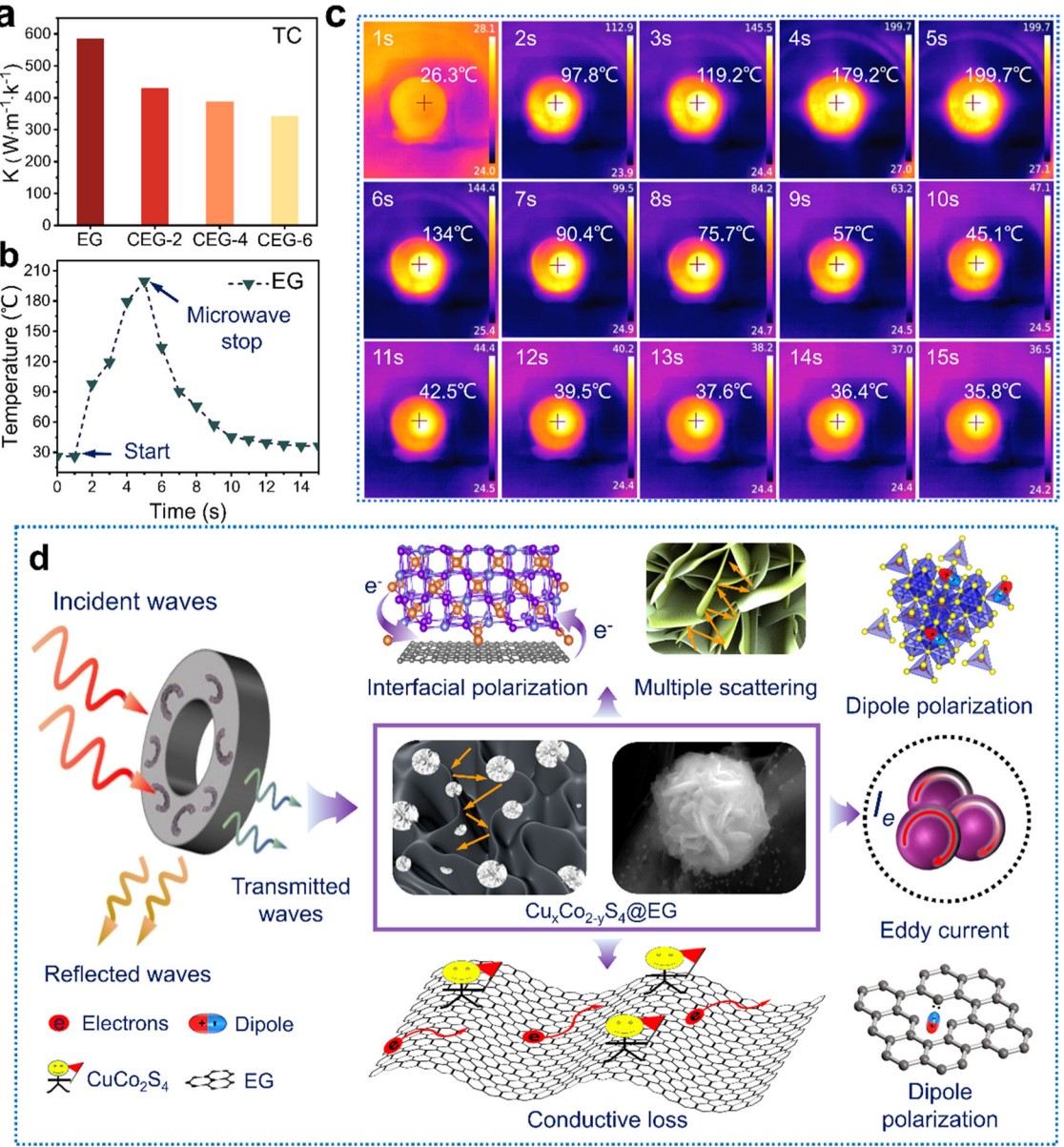

**Fig. 6 | Heat dissipation performance and schematic of electromagnetic wave absorption of CuCo₂S₄@EG. a** Heat dissipation performance of EG, CEG-2, CEG-4 and CEG-6. **b** Heating and cooling curves of CEG-6 at 2.45 GHz microwave frequency. **c** Heating and cooling digital images of CEG-6 under the thermal infrared imager. **d** Schematic of electromagnetic wave absorption of CuCo₂S₄@EG.

constructing the crystal-crystal/amorphous heterointerfaces and cation vacancies. As shown in Fig. 6d, the 3D flower-honeycomb morphology could effectively improve the scattering area of EWM, and reduce the density of the absorber (Supplementary Fig. 31). Meanwhile, this 3D morphology could also function as an interconnected conductive network, offering routes for electron migration and encouraging the creation of conductive losses. Furthermore, the heterointerfaces between EG and CuCo₂S₄ led to uneven distribution of interfacial charge, thereby promoting interfacial polarization. The vacancies and lattice defects present in the EG and CuCo₂S₄ components could result in the generation of dipoles and promote dipole polarization. Therefore, the construction of heterogeneous interfaces by introducing CuCo₂S₄ and the regulation of metal cation defects were effective methods to achieve impedance matching.

In summary, a unique 3D flower-honeycomb CuCo₂S₄@EG composites rich in crystalline-crystalline/crystalline-amorphous interfaces and cation defects were prepared by microwave heating. The component morphology, interfaces and defects of

CuCo₂S₄ were controllable by adjusting the concentration of metal ions. This unique 3D heterostructure and rich cation defects were conducive to promoting interface and dipole polarization, achieving the conduction loss and polarization loss balance, and effectively adjusting the impedance matching of carbon matrix materials. High EMW absorption performance was therefore obtained by achieving multiple attenuation and loss of EMW. Particularly, the EMW absorption performance of CEG-6 was satisfactory, with a RL$_{min}$ of -72.28 dB and an EAB of 4.14 GHz for the composites at a filling ratio of only 7.0 wt. % and a thickness of 1.4 mm. The study showed significant implication about regulating the impedance matching of carbon-matrix absorbing materials through the crystalline-crystalline/crystalline-amorphous heterointerface and cation defects. This work not only proposed a simple and effective strategy for the development and design of high-performance 3D carbon-matrix absorbing materials, but also provided a new perspective for balancing the impedance matching of other EMW absorbing materials.

## Methods

### Materials

All the chemicals used in our experiments, including concentrated sulfuric acid (99%), potassium persulfate (99.9%), copper sulfate pentahydrate ($CuSO_4 \cdot 5H_2O$, 99.8%), cobalt chloride hexahydrate ($CoCl_2 \cdot 6H_2O$, 99.9%), cetyl-trimethyl ammonium bromide (CTAB, 99%), sulfourea (99%), deionized water, ethanol (99.7%), and ethylene glycol (98%), were commercially purchased from Macklin and used without further purification.

### Preparation of 3D honeycomb EG

a high performance carbon-matrix conductive network (EG) was prepared by a microwave-assisted method. First, 2 g of $K_2S_2O_8$ was dissolved in 30 mL of concentrated sulfuric acid, following by adding 0.5 g of graphite. After that, the solution was transferred to a 60 mL of reactor for microwave heating at 60 °C for 5 min, under the conditions of the microwave power of 1200 W, the microwave frequency of $2450 \pm 50$ MHz, and the heating rate of 1 °C/s. After the reaction, the reaction product was repeatedly washed to neutral with distilled water and ethanol, followed by drying. Then the dried reaction products were irradiated by microwave in a quartz crucible for 60 s under the conditions of $Ar_2$ protection and the microwave power of 2000 W. Finally, 3D honeycomb carbon based conductive network (EG) was obtained.

### Synthesis of CuCo₂S₄@EG

the $CuCo_2S_4$@EG heterostructures were prepared by a microwave solvothermal method. Generally, the morphology, interfaces and defects of the composites were regulated by fixing the ratio of S to Co atoms and adjusting the concentrations of Cu and Co atoms. According to different molar ratios (Co:Cu=2:1, 4:1, 6:1, 8:1), a certain amount of $CuSO_4 \cdot 5H_2O$ (0.3 mmol) and $CoCl_2 \cdot 6H_2O$ were completely dissolved in 30 mL of ethylene glycol by ultrasonic stirring. Then a certain amount of thiourea (S: Co=1:1) and 6 g of cetyl-trimethyl ammonium bromide (CTAB) was dissolved in the mixed solution, followed by adding 0.5 g of EG and stirring for 5 min. The final solution was treated in a high-pressure Teflon-lined reaction kettle at 180 °C for 60 min. After cooling, the product was washed and dried to obtain a 3D flower-honeycomb $CuCo_2S_4$@EG heterostructures.

### Characterization

the crystal structure of samples was characterized by XRD (X Pert PRO MPD, PANalytical, Netherlands, Cu Ka), and the Rietveld refinement analysis was utilized to study the crystal structures via TOPAS software. The SEM (SU8030, HITACHI, Japan), HRTEM (TecnaiG2 TF30, FEI, US), and AC-STEM (Theims Z, FEI, US) were employed to analyze the microstructure and morphology. XPS (Thermo Fisher Scientific, UK) were performed to analyze the chemical states of elements. And EPR spectra were recorded from a Bruker EMXplus at 300 K and the frequency of 9.84 GHz. The Cu and Co K-edge X-ray absorption fine structure spectra were collected on the Synchrotron Radiation Facility (easyXAFS 300, easyXAFS, US) under transmittance mode. The DC conductivity ($\sigma$) of the concentric rings shape sample measurements was performed by a semiconductor analyzer (4200A-SCS, Keithley, US). The thermal conductivity of the samples was tested by laser thermal conductivity meter (LFA 467HT Hyperflash, Netzsch, Germany). A vector network analyzer (N5244A PNA, Agilent, US) was used to record the electromagnetic parameters (complex permittivity ($\varepsilon_r = \varepsilon' - j\varepsilon''$) and permeability ($\mu_r = \mu' - j\mu''$)) by a typical coaxial-line method in the frequency range from 1 to 18 GHz. Before detection, all samples (7.0 wt.%) of were compressed into coaxial rings ($\Phi_{in}$= 3.04 mm, $\Phi_{out}$ = 7.00 mm) in paraffin. Accordingly, based on the transmission line theory, the EMW absorption performance and reflection loss (RL) values were given by the following equations:

$$RL = 20\log_{10}\frac{|Z_{in} - Z_0|}{|Z_{in} + Z_0|} \tag{1}$$

$$Z_{in} = Z_0\sqrt{\mu_r/\varepsilon_r}\tanh[j(2\pi f d/c) \times \sqrt{\mu_r\varepsilon_r}] \tag{2}$$

where $Z_0$, $Z_{in}$, d, f and c represent the impedance in free space, the input impedance of the absorber, the thickness of the absorber, and the frequency and velocity of electromagnetic wave of light in vacuum, respectively.

## Data availability

The data supporting the findings of this study are available within the article and the Supplementary Information. Source data are provided with this paper.

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

## Acknowledgements

This work was financially supported by the National Natural Science Foundation of China (Grant Nos. 52374305 and 51864030) (L.X.), and by National Key R&D Program of China (Nos. 2018YFC1901904) (L.X.), and by Yunnan Fundamental Research Projects (Nos. 202301AV070009 and Nos. 202101AS070023) (L.X.), and by Yunnan Provincial Science and Technology Talents Program (Nos. 2019HB003) (L.X.), and by Yunnan Provincial youth top-notch talent support program (L.X.).

## Author contributions

Experimentation, Z.T., C.X.; data curation, Z.T., C.X.; writing – original draft preparation, Z.T., L.X., C.X., L.G.; writing–review & editing, L.X.; methodology, Z.T., L.X., C.X., L.G., L.Z., S.G., and J.P.; investigation, Z.T., L.X., C.X., and L.G.; conceptualization, L.X. and J.P.; supervision, L.X.

## Competing interests

The authors declare no competing interests.
