## [Peer Review File · Nature Communications]

nature portfolio

Peer Review FileReviewer comments, first round

Reviewer #1 (Remarks to the Author):

This manuscript reports the effects of heterogeneous interfaces and defects on the electromagnetic wave (EMW) absorption property of CuCo₂S₄@EG composites. Impressively, the obtained CuCo₂S₄@EG composites exhibit a unique 3D flower-honeycomb morphology, and strong absorption at a very thin thickness and low filler loading. Also, the topic shows specific novelty in related areas. Therefore, this manuscript can be recommended for publication in Nature Communications after some revisions. The manuscript needs to address the following comments:

- (1) What are the advantages of CuCo₂S₄@EG composites used as EMW absorbers in this work? Please give detailed explanations.
- (2) Why did the authors choose the filler loading of 7 wt% for the electromagnetic parameter test?
- (3) BET test should be conducted to further detect the effect of porous structure on the EMW absorption performance of as-prepared CuCo₂S₄@EG composites.
- (4) The authors should provide the PDF cards of CuCo₂S₄ and Cu₃₁S₁₆ in Figure S6 and mark the corresponding crystal planes.
- (5) What causes the notable fluctuation of electromagnetic parameter for EG in Figure 4? Please give detailed explanations.
- (6) Please reorganize and polish the introduction of the manuscript and add related literature in the references to highlight the research background, such as "Journal of Materials Science & Technology 132 (2023) 193-200", "Journal of Colloid and Interface Science 630 (2023) 754-762", "Advanced Electronic Materials 7 (2021) 2001001".
- (7) There are some format errors in the References. Please correct them.
- (8) The paper contains some minor grammatical errors and typo-mistakes that should be corrected.

Reviewer #2 (Remarks to the Author):

This manuscript reported the preparation and electromagnetic wave absorption properties of CuCo₂S₄@EG composites. The interesting results shows that 3D flower-honeycomb morphology, the crystal/amorphous heterointerfaces and the abundant cation defects can effectively adjust the dielectric and magnetic losses, achieve the impedance matching balance and improve the absorption of EMW. The results are characterized and interpreted in detail. However, the manuscript cannot be published at this current stage, it should be substantially revised and the followings should be addressed.

1. The introduction of this paper is too long, need to highlight the focus of this work. The key figures such as TEM are too small and not clear enough in this paper. The lines in the graph describing the electromagnetic parameters are too thick to make obvious changes.
2. In the title "Construction of 3D carbon frame materials through crystal/amorphous heterogeneous interfaces and defects toward electromagnetic wave absorption". Is the "3D carbon frame materials" too broad to give specific information about what this article is studying?
3. It is mentioned that the polarization loss enhances with the increase of Cu/Co ratio, then how does the conductivity loss change? Which mechanism is dominant? For the dielectric loss, the effect of conduction loss is not explained in detail.
4. How are the interface polarization and dipole polarization summarized by the author reflected in the electromagnetic parameters? Is there support material?
5. EG as a pure carbon matrix usually has no magnetic loss, but the complex permeability data of the matrix in the text is different, please explain it. With the introduction of CuCo₂S₄ flower array and the increasing of Cu/Co ratio, Whether it mainly regulates the dielectric loss or magnetic loss of the material? Is there a corresponding explanation.
6. When the concentration of Cu atoms is constant, why the petal-like structure changes with the increase of the number of Co atoms needs further explanation.
7. The CuCo₂S₄@EG heterostructures has more excellent heat dissipation characteristics in C-band compared to the previously reported carbon-matrix absorbing materials. Please provide relevant data support.

8. Balancing the impedance matching usually related to its conductivity. The corresponding change of sample conductivity should be given in this paper.
9. The impedance matching of the material is usually related to its conductivity. In this paper, it is mentioned that the adjustment of heterogeneous interfaces and cation defects can solve the problem of poor impedance matching performance of materials, which needs further explanation.
10. In Figure S3b Raman spectrum, the expansible graphene has a bifurcated peak in the D peak position, please explain.
11. The contribution of interfacial polarization to electromagnetic attenuation has been discussed deeply by peers. Although the authors have made a very detailed characterization and analysis of the polarization mechanism, the research topic of this manuscript lacks highlights. Moreover, the authors don't show how much interfacial polarization contributes to electromagnetic loss.
12. In Figure 4, the corner marker blocks part of the coordinate axis information and needs to be further improved.
13. The corresponding symbols of the real and imaginary parts of the dielectric constant in Figure 4 and Figure S11 should use italics.
14. Some of the pictures in the paper are small and unclear, such as some labels in Figure 3 and Figure 5, please modify them.
15. In page 15 "As demonstrated in Figure 5a1-d1, the CEG-6 sample exhibited the strongest EMW absorbing ability with an ultra-low filler loading (7.0 wt %)". The paper mentions that the load is low, so what is the density of the CEG-6 sample? Does it meet the requirements for low density?

Reviewer #3 (Remarks to the Author):

In this manuscript, the authors constructed a 3D carbon framework composite CuCo₂S₄@EG by designing a heterogeneous interface and cationic defects. Such 3D structure contributes to impedance matching and improves its electromagnetic wave absorption properties in some degree. However, this work lacks novelty in this filed as magnetoelectric composites are very commonly studied. Moreover, many conclusions are unsubstantiated and so many content expressions are confusing with obvious errors. The manuscript cannot meet the high standard of Nature Communications and recommended to be rejected. Some main issues are shown as follows.

1. The morphology of the CuCo₂S₄ is just not the same in different conditions rather than controlled.
2. The statement of "First, the design of the heterointerfaces inherits the unique physical properties of the component materials, such as electrical and magnetic properties and other characteristics." is not correct as intrinsic physical properties have nothing to do with structures.
3. Many conclusions are not convincing or over concluded such as "The impedance balance of the material can be realized to the greatest extent.", "adjust the flower like morphology of CuCo₂S₄ through the proportion of copper and cobalt atoms to enhance the impedance matching of the composites.", "This will lead to lattice mismatch and energy band structure shift, and change the electron distribution state and charge transfer path, thus inducing dipole and interface polarization, and enhancing the dielectric loss of electromagnetic waves" and "which indicates that the defects in CuCo₂S₄ are caused by Cu or Co vacancies (Figure 3h)".
4. The thermal conductivity and electrical conductivity of the material should be compared with the carbon materials prepared by hummers method. (line 84)
5. The conclusion of "the dielectric loss of the composites increases, and it gradually shows a synergistic effect with the magnetic loss" is not specific. What is the synergistic effect?
6. The imaginary part of the complex permeability is less than zero (Fig. 4(e)) which is not an intrinsic property of passive-materials. It should be an artefact caused by improper sample testing, which leads to unreliable results.
7. The impedance matching property cannot be simply described by $Z = |Z_{in}/Z_0|$. This is wrong. There must not be module there. An example is when $Z_{in}=i*Z_0$: Z defined by the authors is unity, but the reflection coefficient is unity as well.
8. The discussion about Cole-Cole plot (Figure S11) cannot be correct as the x-axis and y-axis is not in the identical scale.
9. The labeling in the figures is not clear, especially the text in the illustrations is too small to be legible, e.g., Figure 3 and Figure 5.
10. There are too many mistakes of expressing in the content, such as "construction" should be constructed (line 11), the first sentences of introduction part, "Humer method" (line 82) and "As"

(line 158) etc.

11. The abbreviations of EG, CEG and EAB are missing in the abstract part.

Reviewer #4 (Remarks to the Author):

The paper presents an extensive investigation on expanded graphite-based composite materials, focusing on the microwave absorption properties. Authors expertise in the field is out of doubt, as well as the reliability of the reported experimental results and their potential interest for the skilled reader. Nevertheless I cannot recommend publication by Nature Communications, since a lack of a robust novelty. I suggest to make a deep revision of the text in terms of language and syntax, which are very poor in the present version, and to try to submit the work to other journal more related to carbon-based composites or to microwave absorbing structures. In particular, as hint for increasing novelty, hybrid multilayered composites should be analyzed, at least theoretically, to exploit the different absorbing resonances in order to broaden the EAB: often these kind of articles claim huge EM absorption properties, but it's always a matter of very narrow peaks. Besides, the experimental set up of coaxial waveguide and the sample characterization route is crucial in these kind of sophisticated measurements (and thus should be better described, also by the aid of correspondent images) since the possibility of systematic errors which affect the evaluation of permittivity and permeability, that in turn may critically undermine the calculation of reflection loss as function of thickness (further hint to achieve novelty: once an optimal thickness is advised from the calculation – eqs.1,2 – such finding should be experimentally validated by free space measurement of slabs of the identified thickness, for example by means of NRL Arch method).

Response to Reviewers

We thank for the referee's helpful suggestions. We have incorporated all the referee's suggestions in the revised manuscript. Please view our point-by-point responses below.

Response to reviewer 1

1). What are the advantages of $\text{CuCo}_2\text{S}_4@\text{EG}$ composites used as EMW absorbers in this work? Please give detailed explanations.

R1: Thank the reviewer for carefully reviewing this paper and putting forward professional comments. The three-dimensional (3D) conductive network, porous structure and good stability of EG were conducive to improving charge transport and increasing the propagation path of EMW [1]. EG facilitated achieving strong EMW attenuation at low content when used in a filler matrix for EMW absorption. The morphology and size of double transition metal sulfide with tunable electrical properties were controllable in the synthesis process [2]. Therefore, we have prepared a unique 3D flower-honeycomb $\text{CuCo}_2\text{S}_4@\text{EG}$ composite material rich in crystalline/amorphous heterointerfaces and cationic defects by microwave rapid heating. Meanwhile, the morphology, interfaces and defects of the components were controlled by adjusting the ratio of copper and cobalt atoms. This unique 3D heterogeneous structure can provide abundant heterogeneous interfaces and defects, thereby effectively adjusting the impedance matching of carbon-based materials. And the unique 3D structure enabled EMW inside the material for multiple reflections and refraction, which resulted in multiple absorption and attenuation of EMW. Meanwhile, $\text{CuCo}_2\text{S}_4@\text{EG}$ composites demonstrated excellent properties such as high thermal conductivity, large specific surface area and low density. In addition, the preparation steps of $\text{CuCo}_2\text{S}_4@\text{EG}$ composite material displayed the advantages of simple and short production cycle, high yield, wide source of raw materials, and low cost.

[1] Wang, X. X. et al. Assembling Nano–Microarchitecture for Electromagnetic Absorbers and Smart Devices. *Adv. Mater.* 32, 2002112 (2020).

[2] Li, B. Metal sulfides based composites as promising efficient microwave

absorption materials: A review. *J Mater Sci Technol.* 104, 244 (2022).

2). Why did the authors choose the filler loading of 7 wt% for the electromagnetic parameter test?

R2: Thank the reviewer for your valuable questions. First, we referred to the literature on carbon-based EMW absorbent materials. And it is preliminarily determined that carbon-based materials showed better absorbing properties when the filling amount was about 7 wt.%. [1] At the same time, we also tested the effects of different filling loading (5 wt.% and 9 wt.%) on the absorbing properties of the samples. At the filling loading of 7 wt.%, CEG-6 had the best EMW absorption performance. Its RL_{\min} and EAB were -72.28 dB and 4.14 GHz, respectively (Fig. 1 b-b2). At the filling loading of 9 wt.%, CEG-6 also had good EMW absorption performance. Its RL_{\min} and EAB were -56.56 dB and 4.52 GHz, respectively (Fig. 1 a-a2). However, when the filling loading was 5%, the absorption performance of EMW was poor, and the RL_{\min} and EAB only reached -14.43 dB and 1.58 GHz (Fig. 1 c-c2). In addition, the EMW test results of samples filled at 5 wt.% and 9 wt.% were added to the supporting information.

[1] Liu, Z. C. et al. Self-assembled MoS₂/3D worm-like expanded graphite hybrids for high-efficiency microwave absorption. *Carbon*, 174(2021)59-69.

Fig. 1. EMW absorption performance of CEG-6 at different load amounts, (a-a2) 5.0 wt.%, (b-b2) 7.0 wt.%, (c-c2) 9.0 wt.%

3). BET test should be conducted to further detect the effect of porous structure on the EMW absorption performance of as-prepared $\text{CuCo}_2\text{S}_4@\text{EG}$ composites.

R3: First of all, thank the reviewer for your professional comment. According to the reviewer's suggestion, we tested the specific surface area and pore size distribution of CEG, as shown in Fig. 2. Some micro-medium-large pores were observed for CEG composites. In addition, the specific surface area and pore volume of EG were $34.8 \text{ m}^2\text{g}^{-1}$ and $0.17 \text{ cm}^3\text{g}^{-1}$, respectively. With the introduction of CuCo_2S_4 , the specific surface area and pore volume of the composite decreased. The specific surface area and pore volume of CEG-2 were only $1.1 \text{ m}^2\text{g}^{-1}$ and $0.02 \text{ cm}^3\text{g}^{-1}$. With the change of Co/Cu ratio, the petal-like structure, the specific surface area and pore volume of the composites increased. The specific surface area and pore volume of CEG-6 were $13.7 \text{ m}^2\text{g}^{-1}$ and $0.03 \text{ cm}^3\text{g}^{-1}$. The multi-stage pore structure of CEG-6 composite was beneficial to improve impedance matching [1].

[1] Liu, P. B. Hollow Engineering to Co@N-Doped Carbon Nanocages via Synergistic Protecting-Etching Strategy for Ultrahigh Microwave Absorption. *Adv. Funct. Mater.* 31, 2102812 (2021)

Fig. 2. Pore volume and specific surface area of EG, CEG-2, CEG-4, CEG-6, (a) N₂ adsorption-desorption isotherms, (b) Cumulative pore volume, (c) pore size distribution, (d) The specific surface area

4). The authors should provide the PDF cards of CuCo₂S₄ and Cu₃₁S₁₆ in Figure S6 and mark the corresponding crystal planes.

R4: Thank you very much for the reviewer's advice. According to the comments of reviewer, we marked the PDF card information and corresponding crystal plane of CuCo₂S₄ and Cu₃₁S₁₆ in Fig. 3. It was also revised in the corresponding position of the support information.

Fig. 3. XRD patterns for CuCo₂S₄@EG

5). What causes the notable fluctuation of electromagnetic parameter for EG in Figure 4? Please give detailed explanations.

R5: Thanks to the reviewers for their professional questions. In the range of 1-18 GHz, the dielectric parameter of EG fluctuated obviously, and multiple formants appeared. This was due to conduction loss and polarization relaxation under alternating electric field. For μ' , μ'' , $\tan\delta\mu$ multiple formants were also observed in the range of 1-18 GHz, due to the effect of natural resonance at low frequencies and exchange resonance at high frequencies.

6). Please reorganize and polish the introduction of the manuscript and add related literature in the references to highlight the research background, such as "Journal of Materials Science & Technology 132 (2023) 193-200", "Journal of Colloid and Interface Science 630 (2023) 754-762", "Advanced Electronic Materials 7 (2021) 2001001".

R6: Thank you very much for the reviewer's suggestion. We have rearranged and polished the introduction of the article. We have studied these literatures carefully, which provided good references for the analysis in our paper, and they were quoted in the introduction as references (3, 7and8).

7). There are some format errors in the References. Please correct them.

R7: Thanks to the reviewer, we have carefully checked the format of the reference of the article and corrected the wrong format.

8). The paper contains some minor grammatical errors and typo-mistakes that should be corrected.

R8: We thank for the referee's helpful comments. We have made changes to the language in the article. The modified part of the paper was marked in red.

Response to reviewer 2

1). The introduction of this paper is too long, need to highlight the focus of this work. The key figures such as TEM are too small and not clear enough in this paper. The lines in the graph describing the electromagnetic parameters are too thick to make obvious changes.

R1. Thank you for your helpful comments. We have refined the introduction to highlight the focus of this work. Meanwhile, we have also modified TEM and other key diagrams and lines of electromagnetic parameters according to your suggestions, as shown in Figs. 4 and 5. We have replaced it in the manuscript.

Fig. 4. Microstructure and atomic coordination information of the CEG. (a-b) AC HAADF-STEM images of CEG-6. (c) HAADF-STEM images of CuCo₂S₄. (d) Distribution diagram of CuCo₂S₄, (e) HAADF-STEM images of EG. (f) EPR spectrum of S. (g) Normalized XANES spectra at the Cu K-edge of the Cu foil, CEG-6 and CuO. (h) FT-EXAFS spectra of the Cu foil, CEG-6 and CuO. (i) WT of the Cu foil, CEG-6 and CuO. (j) EXAFS fitting curve for Cu CEG-6 in the k-space. (k) EXAFS fitting curve for Cu CEG-6 in the R-space. (l) Vacancy concentration of CEG samples.

Fig. 5. Electromagnetic wave absorption properties of CuCo₂S₄@EG. (a) Real part and (b) imaginary part of complex permeability. (c) Real part and (d) imaginary part of complex permittivity. (e-f) Dielectric loss tangent ($\tan\delta_\epsilon$).

2). In the title “Construction of 3D carbon frame materials through crystal/amorphous heterogeneous interfaces and defects toward electromagnetic wave absorption”. Is the “3D carbon frame materials” too broad to give specific information about what this article is studying?

R2. Thank you for your important comment. We have modified the title of this paper to show the specific information of this work, and the revised title was as follows: **“Construction of CuCo₂S₄@EG through crystal/amorphous heterointerface and**

defects for electromagnetic wave absorption”

3). It is mentioned that the polarization loss enhances with the increase of Co/Cu ratio, then how does the conductivity loss change? Which mechanism is dominant? For the dielectric loss, the effect of conduction loss is not explained in detail.

R3. Thank you for the professional comment. First, the conductivity of concentric ring samples was measured using a semiconductor parameter analyzer (4200A-SCS, Keithley, USA), as shown in Fig. 6a. Compared to CEG, EG had a higher conductivity (0.74 S.m^{-1}). The conductivity of the samples was significantly lower after the introduction of CuCo_2S_4 compared to EG, and the conductivity of the composite tended to increase with rising Co/Cu ratio.

According to Debye theory, polarization loss (ε_p'') and conduction loss (ε_c'') can be obtained by the following formula [1]:

$$\varepsilon_c'' = \frac{\sigma}{\omega \varepsilon_0} \quad (1)$$

$$\varepsilon_p'' = \frac{\varepsilon_s - \varepsilon_\infty}{1 + \omega^2 \tau^2} \omega \tau = \varepsilon'' - \varepsilon_c'' \quad (2)$$

Where ω is the angular frequency, ε_0 , ε_s and ε_∞ represent the vacuum permittivity ($8.85 \times 10^{-12} \text{ Fm}^{-1}$), the static permittivity and the permittivity at infinite frequencies, respectively, τ refers to the relaxation time and σ is the conductivity.

ε_c'' and ε_p'' of CEG were calculated according to equations (1) and (2), respectively, as shown in Figs. 6b and 6d. The value of ε_c'' for CEG followed the same trend in the frequency range of 1-18 GHz, and ε_p'' increased with the rising Co/Cu. The conduction loss was dominant when it was lower than 2 GHz. The polarization loss was dominant in the range of 2-18 GHz. With the increase of Co/Cu, the polarization loss of CEG increased gradually.

[1] Cao, M. et al. Thermally Driven Transport and Relaxation Switching Self-Powered Electromagnetic Energy Conversion. *Small*. 14, 1800987 (2018).

Fig. 6. (a) DC conductivity of EG versus CEG with 7.0 wt% filler loading, (b-d) The ϵ_c'' and ϵ_p'' of CEG-2, CEG-4 and CEG-6.

4). How are the interface polarization and dipole polarization summarized by the author reflected in the electromagnetic parameters? Is there support material?

R4. Thank you for the professional comment. We have analyzed heterogeneous interfaces and the distribution of cationic defects in CEG composites by AC HAADF-STEM, TEM, XAFS and DFT, and further analyzed the reflection of interface polarization and dipole polarization in the electromagnetic parameters of CEG composites by Debye theory and Cole-Cole curve.

As shown in Fig. 7, the Cole-Cole curve of EG had a long straight section and 9 small semicircles, which were generated by conduction loss and polarization loss, respectively. The mesh skeleton of EG had good conductivity, which was conducive to the generation of conduction loss, while the lattice defects in EG were conducive to the triggering of dipole polarization under the action of electromagnetic wave, resulting in the polarization loss. Compared with EG, the shortening of the linear part of CEG indicated that the conduction loss was reduced. However, the number of semicircles in CEG increased and the shape was distorted. This phenomenon showed

the existence of multiple polarization mechanism. The heterogeneous interface between EG and CuCo_2S_4 and the presence of amorphous-crystal heterogeneous interface in the composite resulted in uneven electron distribution. This promoted the interfacial polarization. In addition, the presence of cationic defects in the CuCo_2S_4 components were favorable for dipole polarization. With the increase of Co/Cu, the concentration of cation vacancies in the CEG was elevated, further promoting polarization loss.

Fig. 7. Cole-Cole plots of (a-a1) EG, (b-b1) CEG-2, CEG-4 and (d-d1) CEG-6

5). EG as a pure carbon matrix usually has no magnetic loss, but the complex permeability data of the matrix in the text is different, please explain it. With the introduction of CuCo_2S_4 flower array and the increasing of Co/Cu ratio,

**Whether it mainly regulates the dielectric loss or magnetic loss of the material?
Is there a corresponding explanation.**

R5. Firstly, thank you for the professional comment. According to the reviewer's comments, we have measured the magnetic properties of EG and CEG composites as shown in Fig. 8. EG showed weak hysteresis. The results displayed the presence of a low coercivity ferromagnetic behavior in the low field region and a significant diamagnetic contribution in the high field region [1,2]. In addition, no obvious hysteresis was observed in CEG composites. Therefore, we mainly regulated the dielectric loss of the composites by introducing CuCo_2S_4 flower-like arrays and varying the Co/Cu ratio.

Fig. 8. (a-b) The hysteresis loop of the CEG and EG

[1] Hota, P. Ferromagnetism in graphene due to charge transfer from atomic Co to graphene. *Appl. Phys. Lett.* 111, 042402 (2017).

[2] Hu, W. Embedding atomic cobalt into graphene lattices to activate room-temperature ferromagnetism. *Nat. Commun.* 12, 1854 (2021).

6). When the concentration of Cu atoms is constant, why the petal-like structure changes with the increase of the number of Co atoms needs further explanation.

R6. Thank you for your careful reviews. During the hydrothermal preparation of CEG, the metal cations (Cu^{2+} and Co^{2+}) have a high affinity for the S^{2-} anion, leading to the formation of CuCo_2S_4 nuclei. Moreover, the high surface energy of CuCo_2S_4 crystal nuclei promoted the two-dimensional growth of CuCo_2S_4 nanoparticles [1]. At the

same time, ethylene glycol and CTAB also played important roles in preventing the agglomeration of individual flakes and promoting homogeneous growth of the nano flower [2-3]. In addition, at higher Co concentrations, Co accelerated the diffusion of Cu, promoting the formation of CuCo_2S_4 sheets [4] and self-assembly into more complex flower-like structures in the presence of CTAB. Based on the reviewer's expert review, we have systematically characterized the formation process for flower-like CuCo_2S_4 , as shown in Fig. 9, and obtained controllable morphologies, which facilitated the refinement of our work.

Figure 9. SEM images under different process conditions, (a)10min, (b)20min, (c)30min, (d)60min, (e)Growth mechanism diagram of CuCo_2S_4 flower

[1] Ge, X. Fabrication of hierarchical iron-containing MnO_2 hollow microspheres assembled by thickness-tunable nanosheets for efficient phosphate removal. *J Mater Chem A*. 4, 14814 (2016).

[2] Liu, P. Facile Synthesis and Hierarchical Assembly of Flowerlike NiO Structures with Enhanced Dielectric and Microwave Absorption Properties. *ACS Appl Mater Interfaces*. 9, 16404 (2017).

[3] Ahmed, S. Unveiling the role of atomic defects on the electronic, mechanical and elemental diffusion properties in CuS. *J Magnes Alloy*. 192, 94-99 (2023).

[4] Shen, B. Morphology Engineering in Multicomponent Hollow Metal Chalcogenide Nanoparticles. *ACS Nano*. 17, 4642 (2023).

7). The CuCo₂S₄@EG heterostructures has more excellent heat dissipation characteristics in C-band compared to the previously reported carbon-matrix absorbing materials. Please provide relevant data support.

R7. Thank you for your helpful suggestions. As shown in Table 1, we have compared the heat dissipation performance of the reported carbon-based EMW absorber materials. In this work, CuCo₂S₄@EG exhibited better heat dissipation performance.

Table 1. Comparison with the heat dissipation properties previously reported for similar carbon materials.

Material	Temperature (°C)	Cooling time (s)	Ref.no.
EG/LLDPE-3D network composites	90 to 20	100	[1]
APPW/EG composites	70 to 20	25	[2]
Co@C/CG aerogels	87.2 to 25.7	15	[3]
EG/BN-103 composites	182.2 to 37.6	16	[4]
EG	83.7 to 36.4	2	[5]
This work	199.7 to 35.8	10	-

[1] Wei, B. J. Polymer Composites with Expanded Graphite Network with Superior Thermal Conductivity and Electromagnetic Interference Shielding Performance. *Chem Eng J*. 404, 126437 (2021).

[2] Xie, Y. P. Highly thermally conductive and superior electromagnetic interference shielding composites via in situ microwave-assisted reduction/exfoliation of expandable graphite. *Compos part A-APPL S*. 149, 106517 (2021).

[3] Xu, J. Lightweight, Fire-Retardant, and Anti-Compressed Honeycombed-Like Carbon Aerogels for Thermal Management and High-Efficiency Electromagnetic Absorbing Properties. *Small*. 17, 2102032 (2021).

[4] Nie, Z. G. Layered-structure N-doped expanded-graphite/boron nitride composites

towards high performance of microwave absorption. *J Mater Sci Technol.* 113, 71 (2022).

[5] Wei, Q. High-performance expanded graphite from flake graphite by microwave-assisted chemical intercalation process. *J Ind Eng Chem.* 122, 562 (2023).

8). Balancing the impedance matching usually related to its conductivity. The corresponding change of sample conductivity should be given in this paper.

R8. Thank you for the professional comment. The conductivity of the samples and the hollow ring EMW absorption test samples were tested using a four-probe powder resistivity tester (ST2253y, Jingge Electronics, China) and a semiconductor parameter analyser (4200A-SCS, Keithley, America), as shown in Fig. 10. It can be seen from the figure that compared with EG, the conductivity of CEG was significantly reduced. However, with the increase of Co/Cu ratio, the conductivity of CEG increased gradually. In addition, we have explored the reason for the increase of CEG conductivity through DFT state density simulation calculation. The result showed that the electronic structure of CEG might be changed by the presence of cationic defects (Fig. 11).

Fig. 10. Conductivity of the EG and the CEG, (a) Pure sample, (b) Sample containing 7 wt.%

Fig.11. (a) The DOS of the CEG heterostructure. (b) The DOS of the CuCo_2S_4 in the CEG, (c) The DOS of the EG in the CEG, (d-f) The PDOS of S, Cu and Co elements in CEG.

9). The impedance matching of the material is usually related to its conductivity. In this paper, it is mentioned that the adjustment of heterogeneous interfaces and cation defects can solve the problem of poor impedance matching performance of materials, which needs further explanation.

R9. Thank for your professional comments. Generally, $|Z_{\text{in}}/Z_0|$ in the regions of 0.8 and 1.2 should be considered as ideal impedance matching performances. The impedance characteristics of different samples were analyzed and characterized. As shown in the Fig. 12, EG had an $|Z_{\text{in}}/Z_0|$ value of only 0.4 and a poor impedance match, which may be caused by its higher conductivity. By in-situ growth of nano flower-like CuCo_2S_4 on the surface of EG, a large number of heterojunctions was formed. The conductivity of the sample was reduced and the $|Z_{\text{in}}/Z_0|$ values were improved. However, the $|Z_{\text{in}}/Z_0|$ values of CEG-2 and CEG-4 were both higher than 1.2, and it was difficult to achieve good impedance matching. When the Co/Cu ratio

was 6:1, the best impedance matching was obtained. This may be due to the fact that modulating the Co/Cu ratio changed the concentration of cation vacancies in the CEG, increasing conduction and polarization losses and regulating the impedance matching of the composite.

Fig. 12. (a₁-d₁) Conduction loss ϵ_c'' and polarization loss ϵ_p'' . (a₂-d₂) $|Z_{in}/Z_0|$ values of EG and CEG. (a₃-d₃) RL maps of EG and CEG.

10). In Figure S3b Raman spectrum, the expansible graphene has a bifurcated peak in the G peak position, please explain.

R10. Thank you for your helpful comments. From the Raman spectrum in Fig. 13, we can observe that the expansible graphite had a bifurcated peak at the position of peak G. It has been shown in the literature that the reduction of symmetry of graphene lamellae in expandable graphite can lead to the splitting of G peak under the action of stress [1,2]. The splitting of G peak of the expandable graphite in this paper was probably due to the stress caused by lattice distortion in the process of hydrothermal

rapid oxidation intercalation.

Fig. 13. The Raman spectra of raw, expansible and expanded graphite

- [1] Ferrari, A. C. Raman Spectrum of Graphene and Graphene Layers. *Phys Rev Lett.* 97, 187401 (2006).
- [2] Frank, O. Compression Behavior of Single-Layer Graphenes. *ACS Nano.* 314, 3131 (2010).

11). The contribution of interfacial polarization to electromagnetic attenuation has been discussed deeply by peers. Although the authors have made a very detailed characterization and analysis of the polarization mechanism, the research topic of this manuscript lacks highlights. Moreover, the authors don't show how much interfacial polarization contributes to electromagnetic loss.

R11. Thank you for your helpful comments, which will help to improve the quality of this work. The significant advantages of heterogeneous interfaces and defect engineering injected unlimited dynamism into the design of advanced carbon-based electromagnetic absorbers. However, how to design heterogeneous interfaces and cationic vacancies to regulate impedance matching and electromagnetic wave (EMW) absorption is an important research field. In this paper, a large number of heterogeneous interfaces were formed by in-situ growth of nano flower-like CuCo₂S₄

on the surface of EG. The conductivity of the sample was reduced, and the impedance values were improved. In addition, the component morphology and defects were regulated by metal ion concentration to achieve an ideal impedance matching. Interestingly, in this paper, flower-like CuCo_2S_4 was grown in situ on the surface of 3D EG. The rich crystalline-crystalline/amorphous heterogeneous interface and the abundance of metal cation defects can effectively adjust conduction loss and polarization loss. It can achieve impedance matching of carbon-based materials, and improve the absorption of EMW. This way was very effective and interesting to prepare high performance absorbing materials. In addition, this work provided a systematic description of the heterogeneous interface, cation vacancy concentration and EMW loss mechanism, which can provide a reference for the development of high performance 3D carbon based materials with application potential.

12). In Figure 4, the corner marker blocks part of the coordinate axis information and needs to be further improved. 13). The corresponding symbols of the real and imaginary parts of the dielectric constant in Figure 4 and Figure S11 should use italics.

R12 and 13. Thanks to the reviewer for pointing out the problems in the figure. We have revised Fig. 4 of the manuscript and Fig. S11 of the supporting materials, as shown in Fig. 14.

Fig. 14. Electromagnetic wave absorption properties of $\text{CuCo}_2\text{S}_4@\text{EG}$. (a) Real part and (b) imaginary part of complex permeability. (c) Real part and (d) imaginary part of complex permittivity. (e) Dielectric loss tangent ($\tan\delta_\epsilon$). (f) ϵ_p'' of CEG. The charge density difference of (g, g₁) heterointerface of $\text{CuCo}_2\text{S}_4@\text{EG}$, (h, h₁) defect CuCo_2S_4 and (i, i₁) nonperfect EG. Blue-green color represents charge depletion, while yellow color represents charge accumulation.

14). Some of the pictures in the paper are small and unclear, such as some labels in Figure 3 and Figure 5, please modify them.

R14. Thanks for the reviewer's suggestions. According to your comments, we have

revised the labels of relevant figures and replaced them in the manuscript.

15). In page 15 “As demonstrated in Figure 5a1-d1, the CEG-6 sample exhibited the strongest EMW absorbing ability with an ultra-low filler loading (7.0 wt %)”. The paper mentions that the load is low, so what is the density of the CEG-6 sample? Does it meet the requirements for low density?

R15. Thank you for the professional comment. The true density of CEG-6 sample was measured by helium displacement method (AccuPyc II 1340, Micromeritics, US) (Fig. 15). The results showed that the true density of CEG-6 was only 0.7020 g/cm³, which satisfied the low density requirement.

Fig. 15. (a-b) True density of CEG-6 samples obtained by helium displacement method

Response to reviewer 3

1). The morphology of the CuCo₂S₄ is just not the same in different conditions rather than controlled.

R1. Thanks to the reviewer for your accurate comments. The morphology of materials had significant influence on electromagnetic wave absorption. Double transition metal sulfides were adjustable in morphology and size during synthesis. We have systematically characterized the morphological changes of CuCo₂S₄ prepared under different technological conditions, as shown in Fig. 16. With the extension of synthesis time, the growth of CuCo₂S₄ petals was promoted. Moreover, the addition

of ethylene glycol and CTAB can prevent the aggregation of single CuCo_2S_4 flakes, and promote the uniform growth of nano petals. In addition, with the increase of Co ion concentration, the synthesized CuCo_2S_4 changed from petal shape to floccus shape. Therefore, the morphology of CuCo_2S_4 can be adjusted by different technological conditions, so as to change the absorption performance of the material.

Fig. 16. SEM images under different process conditions, (a)10min, (b)20min, (c)30min, (d)60min, (e)Growth mechanism diagram of CuCo_2S_4 flower

2). The statement of “First, the design of the heterointerfaces inherits the unique physical properties of the component materials, such as electrical and magnetic properties and other characteristics.” is not correct as intrinsic physical properties have nothing to do with structures.

R2. Thanks very much to the reviewer for your professional comments. The description of this statement was referred to the relevant literature [1]. Meanwhile, based on the reviewer's suggestion and inspiration, we have calculated the density of states (DOS) and the density of fractional states (PDOS) of the cation-rich defective CEG heterostructures by Density functional theory (DFT). This illustrated the

electronic structure changes caused by constructed heterostructures and cationic defects in CEG, as shown in Fig. 17. Compared with CuCo_2S_4 , the band structure of CEG was shifted and the Fermi energy level was reduced. Meanwhile, it can be seen from Fig. 18 that the presence of cationic defects in copper and cobalt formed a new electronic state (p band) near valence band and Fermi level, which was conducive to improving the conductivity of CEG [2].

Fig.17. Calculated DOS of the perfect CEG and CuCo_2S_4 structures

Fig. 18. (a) The DOS of the CEG heterostructure. (b) The DOS of the CuCo_2S_4 in the CEG, (c) The DOS of the EG in the CEG, (d-f) The PDOS of S, Cu and Co elements in CEG.

[1] Liang, L, L, Heterointerface Engineering in Electromagnetic Absorbers: New Insights and Opportunities. *Adv. Mater.* 34, 2106195 (2022).

[2] Zhang, Y. C. Structure-Activity Relationship of Defective Metal-Based Photocatalysts for Water Splitting: Experimental and Theoretical Perspectives. *Adv. Sci.* 6, 1900053 (2019).

3). Many conclusions are not convincing or over concluded such as “The impedance balance of the material can be realized to the greatest extent.”, “adjust the flower like morphology of CuCo_2S_4 through the proportion of copper and cobalt atoms to enhance the impedance matching of the composites.”, “This will lead to lattice mismatch and energy band structure shift, and change the electron distribution state and charge transfer path, thus inducing dipole and interface polarization, and enhancing the dielectric loss of electromagnetic waves” and “which indicates that the defects in CuCo_2S_4 are caused by Cu or Co vacancies (Figure 3h)”.

R3. Firstly, thanks to the reviewers' valuable comments. In order to increase the

accuracy of the description, some conclusions in the paper were carefully demonstrated and revised accordingly.

EG impedance matching was not ideal due to its high conductivity. In this paper, a large number of heterogeneous interfaces were formed by in-situ growth of nano-flower-like CuCo_2S_4 on EG surface. The conductivity of the sample decreased. And the impedance values were improved, but it was difficult to achieve a good impedance matching and ideal EMW absorption performance. Therefore, we hope to achieve better impedance matching by regulating component morphology and defects of CuCo_2S_4 through metal ion concentration, as shown in Fig. 19. It was found that the morphology of CuCo_2S_4 changed from flower-like to flocculent as the Co/Cu ratio was increased. When the Co/Cu ratio was 6:1, the impedance matching of the composite was ideal. This may be due to the change of cationic vacancy concentration in CEG by regulating the Co/Cu ratio and the increase of conduction loss and polarization loss, resulting in better EMW absorption performance of the composites (Fig. 19).

Thanks for the reviewer's comments. The reviewer's questions gave us a great inspiration. We have calculated the density of states (DOS) and the density of fractional states (PDOS) of CEG heterostructures rich in cation defects by Density functional theory (DFT). This can further analyze the effects of heterogeneous structures and cationic defects on changes in the electronic structure of CEG. Compared with CuCo_2S_4 , the band structure of CEG was shifted and the Fermi energy level decreased. In addition, the presence of cationic defects in copper and cobalt formed a new electronic state (p band) near the valence band and Fermi level. This indicated that the construction of cationic defects was conducive to improving the conductivity of CEG. This was consistent with our conductivity test results. At the same time, according to the reviewer's suggestion, we also analyzed the difference of charge density at the non-uniform interface of CEG. The results showed that the charge redistribution at the interface of CuCo_2S_4 and EG resulted in electron accumulation at the interface of CuCo_2S_4 and electron depletion at the interface of EG (Fig. 20a, a1). The uneven distribution of positive and negative charges in space can

promote the interfacial polarization and enhance the dissipation of EMW energy. In addition, we have investigated the charge distribution in the absence of atoms in EG and CuCo_2S_4 , as shown in Fig. 20. The absence of carbon atoms, metal Cu and Co atoms caused the positive charge center to redistribute, creating a local electric field. Finally, a permanent dipole was formed and oscillated in the presence of an external electromagnetic field, enhancing the dipole loss.

In addition, according to the reviewer's comments, we used spherical aberration-corrected transmission electron microscopy (AC-STEM) to collect the microstructure and lattice information of CEG composites. As can be seen from Fig. 21, there were a large number of vacancies in CuCo_2S_4 . To further analyze the coordination environment of Cu and Co atoms, we performed X-ray absorption fine structure (XAFS) measurements of the material (Fig. 21g-k and Fig. 22). The results showed that the coordinations of Cu-S and Co-S in CEG were changed. The coordination number was lower than CuCo_2S_4 (Fd-3m) standard structure. This again indicated that there are two cations missing in CEG, Cu and Co. According to the above analysis, this had an impact on the impedance matching performance of the composite material.

Fig. 19. (a₁-d₁) Conduction loss ϵ_c'' and polarization loss ϵ_p'' . (a₂-d₂) $|Z_{in}/Z_0|$ values of EG and CEG. (a₃-d₃) RL maps of EG and CEG.

Fig. 20. The charge density difference of heterointerface of (a, a₁) CuCo₂S₄@EG, (b, b₁) defect CuCo₂S₄ and (c, c₁) nonperfect EG. Blue-green color represents charge depletion, while yellow color represents charge accumulation

Fig. 21. Microstructure and atomic coordination information of the CEG. (a-b) AC HAADF-STEM images of CEG-6. (c) HAADF-STEM images of CuCo_2S_4 . (d) Distribution diagram of CuCo_2S_4 , (e) HAADF-STEM images of EG. (f) EPR spectrum of S. (g) Normalized XANES spectra at the Cu K-edge of the Cu foil, CEG-6 and CuO. (h) FT-EXAFS spectra of the Cu foil, CEG-6 and CuO. (i) WT of the Cu foil, CEG-6 and CuO. (j) EXAFS fitting curve for Cu CEG-6 in the k-space. (k) EXAFS fitting curve for Cu CEG-6 in the R-space. (l) Vacancy concentration of CEG samples.

Fig. 22. (g) Normalized XANES spectra at the Co K-edge of the Co foil, CEG-6 and Co_3O_4 . (h) FT-EXAFS spectra of the Co foil, CEG-6 and Co_3O_4 . (i) WT of the Co foil, CEG-6 and Co_3O_4 . (j) EXAFS fitting curve for Co CEG-6 in the k-space. (k) EXAFS fitting curve for Co CEG-6 in the R-space. (f-g) Schematic diagram of metal cation vacancy structure of CuCo_2S_4 .

4). The thermal conductivity and electrical conductivity of the material should be compared with the carbon materials prepared by hummers method. (line 84).

R4. Thank the reviewer for your helpful suggestions. The electrical and thermal conductivity properties of materials prepared by hummer method were compared, as shown in Table 2. $\text{CuCo}_2\text{S}_4@\text{EG}$ prepared in this work showed better electrical and thermal conductivity properties.

Table 2. Compared with reported EG and similar carbon materials

Ref.no.	Method	Electrical conductivity	Thermal conductivity
		(s cm ⁻¹)	K (W m ⁻¹ k ⁻¹)
[1]	Oxidation and inserting in HNO ₃ and H ₂ SO ₄ (48-72 h)	300	-
[2]	Oxidation and inserting in HNO ₃ and H ₂ SO ₄ (2-3 days)	386	112
[3]	Oxidation and inserting in H ₂ SO ₄ , H ₂ O ₂ and KMnO ₄ annealing at 800°C for 5 min	-	237
[4]	Sonication in alcoholwater mixture for 20 h annealing at 1060°C for 2 h	850	220
[5]	Chemical vapor deposition	1136	-
[6]	Ball mill with oxalic acid (12 h) and dispersed in NMP annealing at 600°C (2 h)	277	-
[7]	Oxidation and inserting in H ₂ SO ₄ , KMnO ₄ and H ₂ O ₂ (1.35h) annealing at 2000°C for 1h	1000	1100
[8-9]	Oxidation and inserting in H ₂ SO ₄ and HNO ₃ (10-15h) annealing at 800-900°C for 10-20s	1000	-
[10]	Stirring and sonication for 0.5 h, annealing at 450°C (5 min) mechanical compression at 20 MPa (5min)	1467	348
[11]	This work	1012	583

[1] Song, W. L. Magnetic and conductive graphene papers toward thin layers of effective electromagnetic shielding. *J. Mater. Chem. A*. 3, 2097 (2015).

[2] Liang, Q. Z. A Three-Dimensional Vertically Aligned Functionalized Multilayer Graphene Architecture: An Approach for Graphene-Based Thermal Interfacial Materials. *ACS Nano*. 5, 2392 (2011).

[3] Zhao, X. M. Bioinspired modified graphite film with superb mechanical and thermoconductive properties. *Carbon*. 181, 40 (2021).

[4] Hou, Z. L. Flexible Graphene-Graphene Composites of Superior Thermal and Electrical Transport Properties. *ACS Appl. Mater. Interfaces*. 6, 15026 (2014).

- [5] Zhang, L. Preparation and characterization of graphene paper for electromagnetic interference shielding. *Carbon*. 82, 353 (2015).
- [6] Lin, T. Q. Facile and economical exfoliation of graphite for mass production of high-quality graphene sheets. *J. Mater. Chem. A*. 1, 500 (2013).
- [7] Chen, B. Ultrathin Flexible Graphene Film: An Excellent Thermal Conducting Material with Efficient EMI Shielding. *Adv. Funct. Mater.* 24, 4542 (2014).
- [8] Dhakate, S. R. Influence of Expanded Graphite Particle Size on the Properties of Composite Bipolar Plates for Fuel Cell Application. *Energy Fuels*. 23, 934 (2009).
- [9] Dhakate, S. R. Development and Characterization of Expanded Graphite-Based Nanocomposite as Bipolar Plate for Polymer Electrolyte Membrane Fuel Cells (PEMFCs). *Energy Fuels*. 22, 3329 (2008).
- [10] Liu, Y. H. Graphene enhanced flexible expanded graphite film with high electric, thermal conductivities and EMI shielding at low content. *Carbon*. 133, 435 (2018).

5). The conclusion of “the dielectric loss of the composites increases, and it gradually shows a synergistic effect with the magnetic loss” is not specific. What is the synergistic effect?**R5:** Thank the reviewer for your careful and professional review. The expression of this sentence was not accurate enough. According to the reviewer's suggestion, corresponding modifications were made in the original text.

6). The imaginary part of the complex permeability is less than zero (Fig. 4(e)) which is not an intrinsic property of passive-materials. It should be an artefact caused by improper sample testing, which leads to unreliable results.

R6. Thank the reviewer for your professional comments. Some studies have shown that high conductivity of materials may lead to negative imaginary parts of permeability. This may be a phenomenon caused by changes in the electromagnetic field when EMW was transmitted to the surface of a material with higher conductivity. For example, corresponding cases were reported in literature [1-3] (see Figs. 23 and 24). EG showed good electrical conductivity in this paper. In this case, the drop of μ'' to below 0 was consistent with relevant literature reports.

Fig. 23. Electromagnetic parameters of EG and NiCo₂O₄@MnO₂/EG, (a–c) ϵ' , ϵ'' and the dielectric loss tangent, (d–f) μ' , μ'' and the magnetic loss tangent [2]

Fig. 24. Microwave absorption properties of Fe@NCNs, (a) Real part and (b) imaginary part of complex permittivity, (c) Real part and (d) imaginary part of complex permeability, (e) Dielectric loss tangent ($\tan\delta\epsilon$) [3]

[1] Deng, L. Microwave absorbing performances of multiwalled carbon nanotube composites with negative permeability. *Appl. Phys. Lett.* 91, 023119 (2007).

[2] Zhang, X. Honeycomb-like NiCo₂O₄@MnO₂ nanosheets array/3D porous expanded graphite hybrids for high-performance microwave absorber with hydrophobic and flame-retardant functions. *Chem. Eng. J.* 419, 129547 (2021).

[3] Gao, T. Sub-Nanometer Fe Clusters Confined in Carbon Nanocages for Boosting Dielectric Polarization and Broadband Electromagnetic Wave Absorption. *Adv. Funct. Mater.* 2204370 (2022).

7). The impedance matching property cannot be simply described by $Z = |Z_{in}/Z_0|$. This is wrong. There must not be module there. An example is when $Z_{in}=i*Z_0$: Z defined by the authors is unity, but the reflection coefficient is unity as well.

R7. Thank you very much for the valuable comments from the reviewer. For the impedance matching, the modulus of normalized input impedance ($|Z_{in}/Z_0|$) was often selected to characterize the impedance matching condition, which can be expressed by the following formula [1,2].

$$\left| \frac{Z_{in}}{Z_0} \right| = \sqrt{\frac{\mu_r}{\epsilon_r}} \tanh \left[j \left(\frac{2\pi f d}{c} \right) \sqrt{\mu_r \epsilon_r} \right] \quad (1)$$

[1] Gao, T. Sub-Nanometer Fe Clusters Confined in Carbon Nanocages for Boosting Dielectric Polarization and Broadband Electromagnetic Wave Absorption. *Adv. Funct. Mater.* 2204370 (2022).

[2] Qian, J. J. Microwave absorbing performances of multiwalled carbon nanotube composites with negative permeability. *Adv. Eng. Mater.* 2100434 (2021).

8). The discussion about Cole-Cole plot (Figure S11) cannot be correct as the x-axis and y-axis is not in the identical scale.

R8. Thank the reviewer for their professional comments. We have re-evaluated the Cole-Cole curve of composite materials. The Cole-Cole curves of the EG and CEG material were shown in Fig. 25. Compared with EG, the shortening of the linear portion of CEG indicated the reduced conduction loss effect. With the increase of Co/Cu ratio, the number and shape of CEG semicircles were distorted. This implied the phenomenon of multiple polarization.

Fig. 25. Cole-Cole plots of (a-a₁) EG, (b-b₁) CEG-2, (c-c₁) CEG-4 and (d-d₁) CEG-6

9). The labeling in the figures is not clear, especially the text in the illustrations is too small to be legible, e.g., Figure 3 and Figure 5.

R9. Thank the reviewer for pointing out the shortcomings. We have changed the text size in the corresponding graph.

10). There are too many mistakes of expressing in the content, such as “construction” should be constructed (line 11), the first sentences of introduction part, “Humer method” (line 82) and “As” (line 158) etc.

R10. Thank the reviewer for your careful review. We have carefully checked and corrected the grammar and spelling of the article and polished it.

11). The abbreviations of EG, CEG and EAB are missing in the abstract part.

R11. Thank you very much for pointing out the problem. We have made corresponding modifications according to your suggestions.

Response to reviewer 4

1). In particular, as hint for increasing novelty, hybrid multilayered composites should be analyzed, at least theoretically, to exploit the different absorbing resonances in order to broaden the EAB: often these kind of articles claim huge EM absorption properties, but it's always a matter of very narrow peaks. The experimental set up of coaxial waveguide and the sample characterization route is crucial in these kind of sophisticated measurements (and thus should be better described, also by the aid of correspondent images) since the possibility of systematic errors which affect the evaluation of permittivity and permeability, that in turn may critically undermine the calculation of reflection loss as function of thickness (further hint to achieve novelty: once an optimal thickness is advised from the calculation – eqs.1,2 – such finding should be experimentally validated by free space measurement of slabs of the identified thickness, for example by means of NRL Arch method).

R1. First of all, thanks to the reviewers for professional comments and suggestions. In this work 3D flower-honeycomb $\text{CuCo}_2\text{S}_4@EG$ composites rich in crystalline-crystalline/crystalline-amorphous interfaces and cation vacancy was prepared by microwave heating, which the component morphology, interface and vacancy of CuCo_2S_4 can be control by adjusted the concentration of metal ions. This unique 3D heterostructure and rich cationic defects are conducive to promoting interface and dipole polarization, which can achieve the conduction loss and polarization loss balance, and effectively adjust the impedance matching of carbon matrix materials. This will obtain high EMW absorption performance, to achieve multiple attenuation and loss of electromagnetic waves. According to the reviewer's

comments, we reprepared relevant samples and considered using transmission line method (TLM) and NRL-arc method for verification. Therefore, during this period of time, we have built a network analysis platform (E5080B, Keysight, US), retest the previously tested sample (1. Weigh appropriate samples and paraffin wax. 2. Evenly mix the sample with paraffin wax, heat and stir, repeat 3 times. 3. Place the mixture in the mold and press the ring at 10 mpa pressure (Figure 26a). 4. Assemble test parts and load them into samples for test (Figure 26b)). Part of the samples achieved the same test results as before (Figure 26d-i). We also tried to use the NRL-arc method to test the samples. At present, we do not have the test conditions for the NRL-arc method to test, so we also prepared relevant samples and commissioned relevant units to test (Figure 26c). However, due to the large sample size required by the bow method and the complex test parameters, the satisfactory effect has not been achieved.

Fig. 26. (a) samples, (b) network vector analyzer, (c) test the sample by the NRL-arc method, (d-e) EMW absorption performance of CEG-6 method, (g-i) EMW absorption performance of another method.

Reviewer comments, second round

Reviewer #1 (Remarks to the Author):

All comments have been addressed, So it can be accepted.

Reviewer #2 (Remarks to the Author):

I have no other questions. The manuscript can be considered for publication.

Reviewer #4 (Remarks to the Author):

The paper was surely improved in the language and in some technicalities as suggested by the other reviewers. Nevertheless, even stressing again the overall value of the work as well as the reliability of the reported experimental results and their potential interest for the skilled reader, I cannot recommend the paper for publication by Nat. Comm. – due to the journal high standards – for the reasons mentioned at the time (i.e., lack of robust novelty: EAB broadening methods by hybrid design, coaxial waveguide measurements systematic errors analysis, experimental validation by free space techniques, and so on). As previously suggested, however, if not accepted here I warmly encourage the esteemed Authors to submit their valuable work to other journals specifically related to carbon-based composites or to microwave absorbing structures.

Reviewer #5 (Remarks to the Author):

I have noticed that the questions arisen from different reviewers are appropriately replied in the revised manuscript. So I am happy to recommend its accept for publication in Nature Communications.

Response to Reviewers

We thank for the referee's helpful suggestions. Please view our point-by-point responses below.

Response to reviewer 4

1). The paper was surely improved in the language and in some technicalities as suggested by the other reviewers. Nevertheless, even stressing again the overall value of the work as well as the reliability of the reported experimental results and their potential interest for the skilled reader, I cannot recommend the paper for publication by Nat. Comm. – due to the journal high standards – for the reasons mentioned at the time (i.e., lack of robust novelty: EAB broadening methods by hybrid design, coaxial waveguide measurements systematic errors analysis, experimental validation by free space techniques, and so on). As previously suggested, however, if not accepted here I warmly encourage the esteemed Authors to submit their valuable work to other journals specifically related to carbon-based composites or to microwave absorbing structures.

R1: We would like to thank the reviewers for your patience and meticulous review. In this paper, the in-situ growth of flower-like CuCo_2S_4 on the surface of three-dimensional expanded graphite, enriched with crystal-crystal/amorphous heterointerfaces as well as abundant metal cation defects, can effectively regulate the conduction and polarization losses, achieve the impedance matching balance of carbon materials, and improve the absorption of electromagnetic wave. The correlation mechanism of cationic vacancy on material impedance matching was analyzed by means of AC-TEM, XAFS and DFT calculation. This approach to the preparation of high-performance absorbing materials is quite efficient and interesting. In addition, this work systematically describes the effects of heterointerfaces and cation vacancy concentrations on electromagnetic wave absorption, which can provide a reference for the development of high-performance wave-absorbing materials with potential applications.

This work focuses on the construction of 3D flower-honeycomb morphology

CuCo₂S₄@EG composites enriched with crystal-crystal/amorphous heterointerfaces and cationic vacancies, and the modulation of the electromagnetic wave absorption properties of the composites through 3D heterostructures and cationic defects. Of course, we also pay attention to the understanding of different methods for electromagnetic wave performance testing and analysis the generality of different methods for different focuses. Thanks to the reviewer for giving more reference on testing techniques and methods. The experimental testing environment of coaxial-line method, waveguide method and arch method is different, and the results are also quite different. Among them, the coaxial-line method is the most conventional and stable test method for testing the wave-absorbing property of materials. There may be some experimental errors, but the accuracy of measurement can be improved by multiple measurements, thus it has been adopted by most current studies [1-10]. Therefore, our work is also using the coaxial-line method to test the absorption performance, which is also determined by our current test condition. Once again, express our sincerest gratitude.

References

- [1] Iqbal, A. et al. Anomalous absorption of electromagnetic waves by 2D transition metal carbonitride Ti₃CNT_x (MXene). *Science* **369**, 446-450 (2020).
- [2] Cheng, J. Y. et al. Emerging Materials and Designs for Low- and Multi-Band Electromagnetic Wave Absorbers: The Search for Dielectric and Magnetic Synergy? *Adv. Funct. Mater.* **32**, 2200123 (2022).
- [3] Xu, J. et al. Construction of three-dimensional hierarchical porous nitrogen-doped reduced graphene oxide/hollow cobalt ferrite composite aerogels toward highly efficient electromagnetic wave absorption. *J Mater Sci Technol.* **132**, 193–200 (2023).
- [4] Wang, G. H. et al. Integrated multifunctional macrostructures for electromagnetic wave absorption and shielding. *J. Mater. Chem. A.* **8**, 24368-24387 (2020).
- [5] Wang, X. X. et al. Assembling Nano–Microarchitecture for Electromagnetic Absorbers and Smart Devices. *Adv. Mater.* **32**, 2002112 (2020).

- [6] Huang, Y. J. et al. Catenary Electromagnetics for Ultra-Broadband Lightweight Absorbers and Large-Scale Flat Antennas. *Adv. Sci.* **6**, 1801691 (2019).
- [7] Shu, R. W. et al. Synthesis of FeCoNi/C decorated graphene composites derived from trimetallic metal-organic framework as ultrathin and high-performance electromagnetic wave absorbers. *J Colloid Interf Sci.* **630**, 754–762 (2023).
- [8] Shu, R. W. et al. Nitrogen-Doping-Regulated Electromagnetic Wave Absorption Properties of Ultralight Three-Dimensional Porous Reduced Graphene Oxide Aerogels. *Adv. Electron. Mater.* **7**, 2001001 (2021).
- [9] Zhang, X. et al. Honeycomb-like NiCo₂O₄@MnO₂ nanosheets array/3D porous expanded graphite hybrids for high-performance microwave absorber with hydrophobic and flame-retardant functions. *Chem. Eng. J.* **419**, 129547 (2021).
- [10] Xie, Y. D. et al. Ultra-Broadband Strong Electromagnetic Interference Shielding with Ferromagnetic Graphene Quartz Fabric, *Adv. Mater.* **34**, 2202982 (2022).